# ARTS, the atmospheric radiative transfer simulator — version 2.2, the planetary toolbox edition

Stefan A. Buehler[1], Jana Mendrok[2], Patrick Eriksson[2], Agnès Perrin[4], Richard Larsson[5], and Oliver Lemke[1]

[1]Meteorologisches Institut, Centrum für Erdsystem- und Nachhaltigkeitsforschung (CEN), Fachbereich Geowissenschaften, Fakultät für Mathematik, Informatik und Naturwissenschaften, Universität Hamburg, Bundesstrasse 55, 20146 Hamburg, Germany
[2]Department of Space, Earth and Environment, Chalmers University of Technology, SE-41296 Gothenburg, Sweden
[4]Laboratoire de Météorologie Dynamique/IPSL, UMR CNRS 8539, Ecole Polytechnique, Université Paris-Saclay, RD36, 91128 PALAISEAU Cedex, France
[5]National Institute of Information and Communications Technology, 4 Chome-2-1 Nukui Kitamachi, Koganei, Tokyo JP-184-0015, Japan

*Correspondence to:* S. A. Buehler (stefan.buehler@uni-hamburg.de)

**Abstract.** This article describes the latest stable release (version 2.2) of the Atmospheric Radiative Transfer Simulator (ARTS), a public domain software for radiative transfer simulations in the thermal spectral range (microwave to infrared). The main feature of this release is a planetary toolbox, that allows simulations for the planets Venus, Mars, and Jupiter, in addition to Earth. This required considerable model adaptations, most notably in the area of gaseous absorption calculations. Other new

features are also described, notably radio link budgets (including the effect of Faraday rotation that changes the polarisation state), and the treatment of Zeeman splitting for oxygen spectral lines. The latter is for example relevant for the various operational microwave satellite temperature sensors of the Advanced Microwave Sounding Unit (AMSU) family.

## 1 Introduction

Numerical radiative transfer (RT) modeling with computers perhaps started from the urge to understand atmospheric radiant

energy fluxes. The earliest general circulation model (Phillips, 1956) did not yet include a radiation scheme, but simply assumed a globally constant radiative heating rate. In the same year, Plass (1956) already published an article describing numerical simulations of infrared radiation. This paved the way for simple one-dimensional radiative convective models of Earth's energy balance (Manabe and Möller, 1961), and later for global circulation models with sophisticated radiation schemes. It is fair to say that numerical radiative transfer simulations started as soon as computers were becoming available to atmospheric scientists.

Since then, the atmospheric sciences have had a constant need for ever more accurate and efficient RT simulation software.

Besides radiative energy flux calculation, the other important application area for RT software is remote sensing. This started almost at the same time as the energy flux simulations, an early example is Kaplan (1959). From the early days on, high-level computer codes for energy flux computation and remote sensing simulations have developed somewhat independently, and not many complex codes can be used for both applications. Notable exceptions are libRadtran (Emde et al., 2016), which

can be used for sensor simulation and flux calculation in the shortwave, and the family of models by AER (Atmospheric and Environmental Research, Clough et al., 2005). The tendency for models to specialise is often not driven by physics (for example low level solvers like DISORT (Stamnes et al., 1988) are suitable for both applications), it rather seems to be driven by practical constraints, resulting from the requirements of the two communities.

A similar partitioning exists even among the remote sensing RT codes themselves. Historically, most codes were developed for a particular sensor, or remote sensing technique, so that there are dedicated codes for active or passive sensors, microwave, infrared, or ultraviolet/visible frequencies, and up-looking, down-looking, or limb-looking geometry. Moreover, such partitioning also exists regarding the object of observation like the different bodies of the solar system.

Radiative transfer models for planets other than Earth have been developed about equally as long as for Earth itself (e.g.,
Cess, 1971). Also, terrestrial radiative transfer codes have frequently been used to simulate spectra of solar system as well as exo-planets with certain modifications or extensions of, e.g., the spectroscopic data applied (e.g. Urban et al., 2005; Bernstein et al., 2007; Kasai et al., 2012; Vasquez et al., 2013a, b; Schreier et al., 2014). Few have been explicitly developed with a view on applicability to a wide range of different planet characteristics, like e.g., VSTAR (Versatile Software for Transfer of Atmospheric Radiation, Bailey and Kedziora-Chudczer, 2012) or SMART (Spectral Mapping Atmospheric Radiative Transfer,
Meadows and Crisp, 1996). Interest in prediction and analysis of non-Earth spectra has increased significantly in recent years due to intensified research into habitability of planets and the search for exoplanets, calling also for more consistent and more generally applicable models.

Regarding Earth observations, the separate development of models for spectral regions or measurement techniques now proves to be an obstacle for synergistic use of modern multi-sensor observations, which requires consistency in the simulation
of all involved sensors. Out of an appreciation of this, a few RT codes have been developed that are fairly broad in scope, agnostic of a particular sensor, and used for a wide range of applications. Besides the already mentioned AER model family (Clough et al., 2005) and libRadtran (Emde et al., 2016), Dudhia (2017) and Schreier et al. (2014) could be named here as general-purpose models for the infrared spectral range; and of course the Atmospheric Radiative Transfer Simulator (ARTS), the subject of this article.

The ARTS project started in the year 2000 as a joint initiative of Patrick Eriksson (Chalmers) and Stefan Buehler (then at University of Bremen). Table 1 presents a very brief summary of general ARTS features. Right from the start the code was open source (GNU's Not Unix (GNU) public license); the current version is freely available at www.radiativetransfer.org. At the start, the model focused on simulating clear-sky limb observations of Earth's atmosphere in the millimeter and sub-millimeter spectral range, because that was the main interest of the authors (Eriksson et al., 2002; Buehler et al., 2005b). Pretty
soon, the interests widened, and ARTS adopted new capabilities such as simulating downlooking meteorological microwave sensors (Buehler et al., 2004; John and Buehler, 2004) and active radio link measurements (Eriksson et al., 2003). ARTS was also started to be used for infrared energy flux simulations (Buehler et al., 2006b; John et al., 2006), and the capability to handle cases with scattering by hydrometeors was developed by two different scattering solvers, the discrete ordinate iterative solver (DOIT, Emde et al., 2004a, b), employed for example in Rydberg et al. (2007) and Sreerekha et al. (2008), and a Monte
Carlo solver (MC, Davis et al., 2005, 2007), for example used in Rydberg et al. (2009) and Eriksson et al. (2011d).

**Table 1.** An overview of general ARTS features.

| | |
|---|---|
| Name | Atmospheric Radiative Transfer Simulator (ARTS) |
| Website | radiativetransfer.org |
| Programming language | C++ (with accompanying tools in Python and Matlab) |
| Flow control | Scripting-language-like controlfiles allow large flexibility in calculation setup |
| Input and output file formats | XML, NetCDF, some specialised formats for spectroscopic data (e.g., HITRAN) |
| License | GNU Public License |
| Absorption calculation types | Line-by-line or lookup table (absorbing species see Table 4) |
| Spectral range for absorption calculation | Microwave to visible |
| Spectroscopic data | Data up to 3 THz are included for Earth, Venus, Mars, and Jupiter; standard databases (e.g., HITRAN) can be used at higher frequencies |
| Continuum absorption | Built-in continuum absorption models for microwave to infrared (but not visible) |
| Radiative transfer calculation type | Solves monochromatic pencil beam radiative transfer equation with thermal emission and optional scattering, pure transmission calculation also possible |
| Source function | Planck function or pure extinction (using physical temperature as source function for Rayleigh-Jeans limit calculations also works, but is not recommended) |
| Spectral range for radiative transfer simulation | Microwave to thermal infrared (no collimated beam solar source) |
| Viewing geometries | Up-looking, down-looking, limb-looking, sensor inside or outside the atmosphere |
| Model geometry | Spherical 1D, 2D, or 3D (with plane parallel as limiting case for large planet radius) |
| Polarisation | Scalar intensity, selected Stokes components, or full Stokes vector |
| Surface roughness | Specular reflection or arbitrary reflection pattern |
| Surface topography | Allowed for 2D and 3D geometry, none for 1D by definition |
| Passive sensors | Comprehensive linearised sensor treatment for efficient weighting of monochromatic pencil beam radiances |
| Active sensors | Radio occultation (intensity only, no wave propagation) |
| Scattering solvers | Discrete Ordinate Iterative (DOIT) solver; Monte Carlo (MC) solver (for the stable version described in this article, the development version includes several additional solvers) |
| Single scattering data | Absorption vector, extinction matrix, and discrete angular grid 4x4 phase matrix (have to be externally generated) |
| Jacobian calculation | Analytical and/or semi-analytical for clear-sky variables; no Jacobians in the presence of scattering in the version described in this article, but this feature is under development |

ARTS comes with a quite complete set of documentation, consisting of four main elements: First, the top level directory of the distribution contains several readme files that describe the program configuration, compilation, and execution. Command line options are also explained by the program itself when run with the '-h' or '–help' command line option. Configuration and compilation follow standard open source unix programming conventions. Second, there are the guide books (User Guide, Theory Guide, and Developer Guide), which give a comprehensive overview of the program from a user perspective, from a theoretical perspective, and from a programming perspective, respectively. Third, ARTS works like a scripting language with functions (in ARTS called *methods*) that work on variables (in ARTS called *workspace variables*), and each of these functions and variables has built-in documentation, perhaps comparable to a Unix man page, that can be browsed online at the ARTS website. Fourth, the distribution includes a large set of sample *controlfiles* for ARTS that contain predefined setups for various remote sensing instruments, and demonstration cases for various ARTS features. There also is a build target 'make check' that runs a selection of the included controlfiles and compares their computation results against reference data. For the user, this allows to verify that the model works correctly. For the developer, perhaps even more importantly, it helps to ensure continuity and prevents unintentional changes in model output due to source code changes.

Over the years, the model was validated by several inter-comparison studies (e.g., Melsheimer et al., 2005; Buehler et al., 2006a; Schreier et al., 2018). Quite recently, the ARTS infrared energy flux calculations were used as one of the reference models in a broad assessment of the quality of radiation codes in climate models (Pincus et al., 2015), and were shown to be in very good agreement with the other participating reference models. Also, closure studies with radiosondes, microwave observations, and infrared observations increase our confidence that the model consistently handles the different spectral ranges (Kottayil et al., 2012; Bobryshev et al., 2018).

Perhaps the most significant limitation, though, that remains even to date, is that ARTS does not have a collimated beam source, so it currently cannot simulate solar radiation observations or solar radiation energy fluxes. The line-by-line absorption calculation itself, however, does work also in the solar spectral range, and has been used by Gasteiger et al. (2014) to precalculate absorption cross sections for libRadtran (Emde et al., 2016), using the simulated annealing method described in Buehler et al. (2010).

There are only two previous publications that describe earlier versions of ARTS as a whole, Buehler et al. (2005a) and Eriksson et al. (2011a), but many of the main building blocks of ARTS and the tools around it have been described in dedicated publications. Besides the already mentioned DOIT and MC scattering solvers, important building blocks are the method to pre-calculate and store gas absorption data (Buehler et al., 2011) and the method to handle sensor characteristics by building up a comprehensive sparse matrix sensor representation (Eriksson et al., 2006).

Important tools around ARTS are the Qpack Matlab package (Eriksson et al., 2005) that, among many other things, allows optimal estimation inversions (going from measured or simulated radiation back to an estimate of the atmospheric state), and a Matlab package for frequency grid optimisation by simulated annealing (Buehler et al., 2010), which both are part of the bigger Matlab package ATMLAB (ATMospheric matLAB), freely available from the ARTS website. The website also holds arts-xml-data, a data package with model atmospheres, spectroscopic data, and other data that are required or useful for running

radiative transfer simulations. And, last but not least, there is a growing set of Python interface and helper functions, collected in a package called Typhon.

This article describes ARTS version 2.2. The most visible difference to prior versions is that the program, originally developed for Earth, has been adapted to also work well for the other solar system planets, specifically Mars, Venus, and Jupiter. These additions were developed in a study supported by the European Space Agency (ESA). Along with the program itself comes a set of inputs for the different planets, such as spectroscopic parameters, atmospheric composition, and basic parameter settings such as the planet's radius. Together, program and input data form what we call the planetary toolbox.

Details on the input data and the actual performance of the model relative to planetary observations will be the subject of another planned article, but to advertise the capability, Figure 1 shows simulations of space-based nadir observations of the 100-300 GHz spectral region for the four different planets. Quite different molecular species dominate this spectral region for the different planets: $SO_2$ and $H_2SO_4$ spectral features on a background of collision-induced $CO_2$ absorption for Venus, and prominent $O_2$ and $H_2O$ lines with some minor $O_3$ features for Earth. For Mars, one mostly sees the surface, with some very narrow emission lines ($H_2O$, $CO$), due to the very thin atmosphere. For Jupiter, the most prominent feature is a strong $PH_3$ line, that sits on an absorption background due to $NH_3$, modulated by several broad $H_2S$ absorption features.

There are some caveats for the spectra in Figure 1. First of all, shown are nadir brightness temperatures, which should be kept in mind when comparing to disk integrated measured brightness temperatures. Second, these are clear-sky simulations, neglecting the influence of cloud or precipitation particles, which may affect observed spectra. Third, especially the $NH_3$ absorption for Jupiter has been shown to be highly sensitive to the choice of spectral line shape (Encrenaz and Moreno, 2002); we have used a Voigt shape. Last, of course, spectra may differ also strongly for other atmospheric scenarios.

Besides the planetary toolbox, there were numerous other additions and improvements: To start with, ARTS now includes collision induced absorption continua from the HIgh-resolution TRANsmission molecular absorption database (HITRAN, Richard et al., 2012). This addition was motivated by the urgent need for some of these continua for other planets, but they may be useful for Earth as well.

Another change is, that the program generally has far fewer internal constants now, which instead are read from input files; or that rather can be read, because there are still built-in default values for convenience. This applies for example to isotopologue ratios and to spectroscopic partition functions. Also in the area of spectroscopy, pressure broadening has been generalised to use separate broadening parameters for all major broadening gas species of the different planets.

Capabilities to simulate active observations have been enhanced by correctly treating Faraday rotation for radio links. The implementation of this effect uses a Stokes vector formalism where the extinction term in the scalar radiative transfer equation is replaced by a four-by-four propagation matrix. This has benefited greatly from the experience gathered with the last important addition that has to be mentioned here: the capability to simulate oxygen Zeeman splitting in a physically rigorous way, which is also handled by a Stokes vector formalism, described in Larsson et al. (2014); Larsson (2014). The method has been validated against observations in uplooking (Navas-Guzmán et al., 2015) and downlooking (Larsson et al., 2016) geometry, and also has already been employed for some sensitivity and retrieval simulation studies (Larsson et al., 2013, 2017).

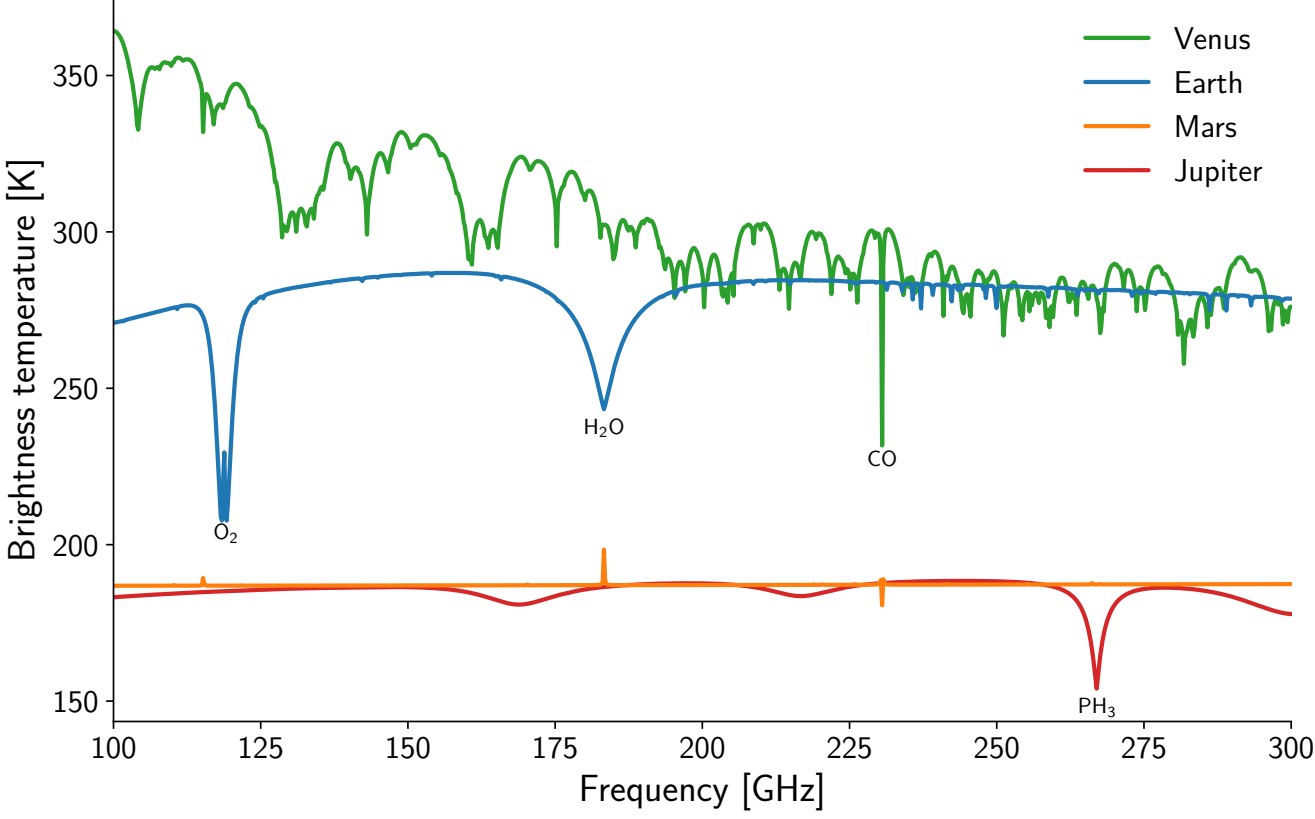

**Figure 1.** Simulated millimeter-wave nadir observations from space for four different planets. The atmospheric scenarios are the same as the ones behind Table 2 (vertical profiles, although the table just lists the data for a single pressure level). Surface reflectivity was assumed to be 0.4 for Earth, 0.13 for Mars, and does not play a role for the other two planets. The large spectral variability for Venus is caused by $SO_2$.

The main purpose of this article is to introduce, explain, and document these recent extensions and modifications, and to serve as a reference for this version of ARTS. The structure is as follows: Section 2 describes the planetary toolbox extensions and modifications, Section 3 describes other modifications and extensions, and Section 4 contains summary and outlook.

## 2    From Earth to planets: generalized propagation modeling methods

5   When extending radiative transfer modelling from Earth to other planets, the major challenge is to remove a number of assumptions on basic physical parameters made in the model itself or in the input data. Issues include hard-coded constants that are valid (and constant) for Earth, but might differ between planets. They furthermore include assumptions in certain algorithms and parameterisations. The most prominent one is the expression of spectroscopic parameters of gas absorption lines like foreign pressure broadening and pressure induced frequency shifts by a single parameter valid for the standard mixture of

**Table 2.** Basic composition of different planets. The table lists VMR values at 700 Pa for some basic model atmospheres that are distributed with ARTS as part of the 'arts-xml-data' package. (See documentation in arts-xml-data for data origin, the used scenarios are: Venus: Venus.vira.day; Earth: Fascod/tropical; Mars: Ls0.day.dust-medium.sol-avg; Jupiter: Jupiter.mean.) The last row, '$T$' lists temperature values from the same atmospheres. The last column, '$\gamma$-183', lists pressure broadening parameters of the 183-GHz $H_2O$ line in kHz/Pa. Note that the VMRs are not normalized, so they do not exactly add up to 1 for each planet. Also note that these are only the gases for which we have dedicated broadening parameters; all planets also have other tracegases that are spectroscopically active.

|        | Venus  | Earth  | Mars   | Jupiter | $\gamma$-183 |
|--------|--------|--------|--------|---------|--------------|
| $N_2$  | 4.4%   | 78%    | 2.7%   |         | 31           |
| $O_2$  | 5.1e-7 | 21%    | 9.7e-4 |         | 20           |
| $H_2O$ | 6.2e-7 | 4.5e-6 | 1.5e-4 | 5.0e-11 | 155          |
| $CO_2$ | 97%    | 3.3e-4 | 95%    | 3.9e-12 | 51           |
| $H_2$  |        |        | 1.0e-5 | 86%     | 24           |
| He     |        |        |        | 14%     | 7            |
| $T$    | 203    | 241    | 204    | 155     |              |

air (79% $N_2$+21% $O_2$). Here, the limitation is not only in the RT model itself, but also in the spectroscopic catalogues, which commonly report the Earth-valid standard air parameters only.

ARTS has been revised for such assumptions, and modifications towards more general approaches have been made. Below we detail the most relevant of them.

## 2.1 Line spectroscopy

Spectral lines are broadened by collision of gas molecules with other gas molecules. The line width then scales with the partial pressure of the perturbing species. The constant of proportionality is specific to each transition and to the species involved.

Commonly in line-by-line absorption modeling, self broadening and foreign or air broadening are distinguished. The total line width is the sum of the self broadened line width and the foreign broadened line width. The self broadened line width scales with the partial pressure of the species itself, while the foreign broadened line width scales with the total pressure minus the partial pressure of the species itself. (For an explicit mathematical formulation, see Equation 1 further down.) It is typically the respective broadening proportionality constants, or broadening coefficients, of self and foreign broadening, which are reported in the spectral line catalogues.

For line catalogues focusing on Earth applications, the reported foreign broadening coefficient is derived for a standard air mixture of 79% $N_2$ and 21% $O_2$. When considering other planets than Earth, the assumption of air as a nitrogen-oxygen-mixture does not hold anymore. Instead, the composition of the atmosphere varies hugely from planet to planet, as illustrated by the example atmospheric compositions shown in Table 2.

Pressure broadening is specific to the species involved, and this is also illustrated in Table 2, for the example of the 183 GHz water vapor line. Since atmospheric composition affects the pressure broadening, the true composition must be considered for

exact calculations. The consequences of not calculating the broadening correctly can be drastic: In a recent comment, Turbet and Tran (2017) point out that using air instead of the correct $CO_2$ broadening coefficients may lead to an error of 13 K in the surface temperature in climate simulations for early Mars.

In principle, the impact of the basic atmospheric composition of another planet on the line broadening can be and often is handled in the way that the concept of a foreign broadening coefficient given for a standard air mixture is kept. This requires the compilation of spectral line catalogues specific to the atmospheric composition in question, i.e., the compilation of catalogues specific to individual planets. A more flexible option, though, is to explicitly report broadening parameters for the variety of broadening gases in the line catalogue and derive the foreign broadening coefficient from them just-in-time considering the actual atmospheric composition. The latter approach has been chosen for ARTS.

In addition to the line broadening, gas molecule collisions cause pressure dependent frequency shifts of the transitions, also called pressure shifts. Just as the broadening, the pressure shifts are specific to each transition and the species involved in the collision. Commonly, only an overall pressure shift parameter is reported in line catalogues and applied in the line-by-line absorption modeling. Regarding applicability in atmospheres of different compositions, similar considerations as presented for line broadening apply to pressure shifts.

Earlier ARTS versions (Buehler et al., 2005a; Eriksson et al., 2011a) follow the common approach of standard air foreign broadening and pressure shift parameters, calculating the pressure broadened line width $\gamma_L$ as

$$\gamma_L = x_{\rm s}\, p\, \gamma_{\rm s}\, \left(\frac{T_{\rm ref}}{T}\right)^{n_{\rm s}} + (1 - x_{\rm s})\, p\, \gamma_{\rm a}\, \left(\frac{T_{\rm ref}}{T}\right)^{n_{\rm a}} , \tag{1}$$

where the first term on the right hand side denotes the self broadening width $\gamma_{L{\rm s}}$ and the second one the foreign or air broadening width $\gamma_{L{\rm a}}$. In Equation 1, $\gamma_{\rm s}$ and $\gamma_{\rm a}$ are the self and the air broadening parameters, $n_{\rm s}$ and $n_{\rm a}$ are the temperature exponents for $\gamma_{\rm s}$ and $\gamma_{\rm a}$, respectively, and $T_{\rm ref}$ is the reference temperature of the broadening parameters. All these parameters are reported in spectroscopic catalogues (the reference temperature often only implicitly for the entire catalogue). Furthermore, $x_{\rm s}$ is the volume mixing ratio (VMR) of the transition species, $p$ is the total atmospheric pressure and $T$ is the atmospheric temperature.

The pressure shift $\Delta\nu$ is calculated as

$$\Delta\nu = p\, \delta\nu\, \left(\frac{T_{\rm ref}}{T}\right)^{(0.25 + 1.5*n_{\rm a})} , \tag{2}$$

where $\delta\nu$ is the pressure shift parameter reported in spectroscopic catalogues. Note that to our knowledge there is no generally accepted formulation for the temperature dependence of $\Delta\nu$ and that Equation 2 simply reports the expression applied in ARTS, without any claim of general validity. The origin of these values for our model is in Pumphrey and Buehler (2000), which in turn refers to Pickett (1980), but that paper, although it does discuss the theory of the pressure shift temperature dependence, does not give any explicit value suggestions for the exponents. Despite its shortcomings, we decided to keep the expression for continuity, and in lack of a better one.

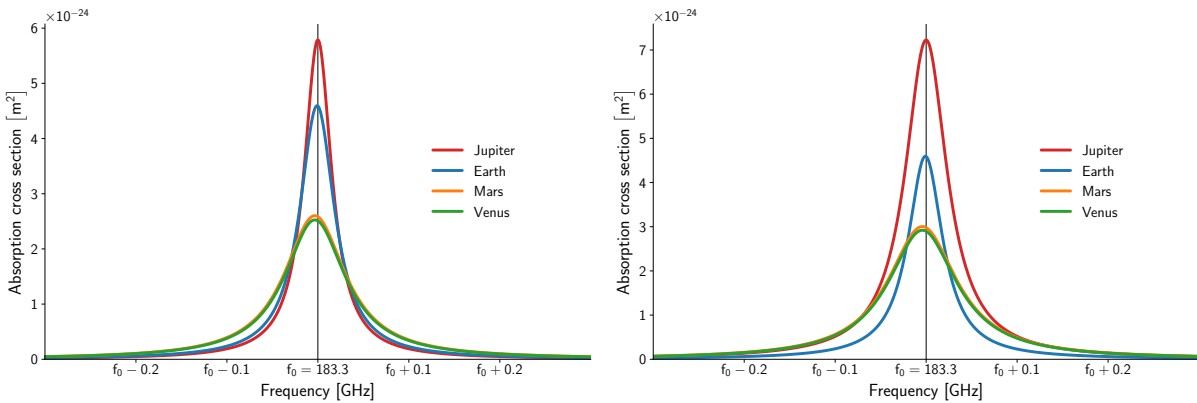

**Figure 2.** Absorption cross section of the 183 GHz water vapor line at 700 Pa, for the different atmospheric compositions listed in Table 2, assuming a Voigt line shape function. Left: using the Earth atmosphere temperature for all four cases, so that differences are only due to the different pressure broadening coefficients. Right: Taking also the temperature from the different planet scenarios, which affects the line strength, corresponding to the integral under the curves.

To allow for flexible air compositions, the foreign broadening width $\gamma_{La}$ has been reformulated into a weighted sum of the broadening contributions from individual broadening species as

$$\gamma_{La} = (1 - x_s)\, p\, \frac{\sum_i \left[ x_i\, \gamma_i\, \left( \frac{T_{ref}}{T} \right)^{n_i} \right]}{\sum_i x_i} \, , \tag{3}$$

where $\gamma_i$ is the broadening parameter of the $i^{th}$ broadening species, $n_i$ its temperature coefficient, and $x_i$ the VMR of the broadening species. To illustrate the impact of this new treatment, Figure 2 shows the absorption cross section of the same water vapor line in the atmosphere of four different planets.

Similarly, pressure shift $\Delta\nu$ has been rewritten as

$$\Delta\nu = p\, \frac{\sum_i \left[ x_i\, \delta\nu_i\, \left( \frac{T_{ref}}{T} \right)^{(0.25 + 1.5 * n_i)} \right]}{\sum_i x_i} \, , \tag{4}$$

with $\delta\nu_i$ being the pressure shift due to the $i^{th}$ broadening, or rather shifting, species. Note that, like Equation 2, Equation 4 simply states the formula used in ARTS, without claiming general validity. The shift effect can also seen in Figure 2, if one looks closely the peaks of the cross section curves for Venus and Mars are noticeably different from those of Earth and Mars.

Commonly, the atmospheric composition is not specified in such detail that the sum over the VMR of all considered species adds up to 1. For the classical approach, Equations 1 and 2, this does not matter as the contribution from all foreign gases is taken into account by weighting the foreign contribution with the total foreign air pressure $((1 - x_s)\, p)$. For the revised approach, Equations 3 and 4, the normalisation by $\sum_i x_i$ balances out deviations from a VMR sum of 1.

A completely general approach would have to take into account $\gamma_i$ and $n_i$ for all possible atmospheric gas species $i$. Since contributions of individual species scale with their VMR, it is sufficient to cover the major atmospheric gas species and neglect

minor trace gas species. Currently, ARTS 2.2 considers $N_2$, $O_2$, $H_2O$, $CO_2$, $H_2$, and He as foreign broadening species. This selection covers the most abundant species in the atmospheres of Venus, Earth, Mars, and Jupiter, the planets the toolbox has been developed for. The approach itself is generally applicable, and the ARTS implementation could easily be modified to cover further foreign species.

This new broadening mechanism has theoretical advantages even for Earth's atmosphere. To give an example, the broadening of oxygen lines by water vapor is stronger than their nitrogen broadening, which makes oxygen lines broader in a very wet atmosphere. So far, it was not possible to treat this effect with a generic line-by-line calculation based on an external catalog, but with the species-specific broadening parameters in the new ARTS catalogue it happens automatically, if parameters for the broadening by water vapor are available.

However, the practical difference that this makes for Earth is really small. For the example of the 119 GHz oxygen line, the water vapor broadening parameter is roughly 12% larger than the nitrogen broadening parameter (and the oxygen or self-broadening parameter is quite similar to the nitrogen one). Assuming a water vapor VMR of 1% then increases the total width of the line by only about 0.13%. The reason for the small impact is that there is so much more nitrogen and oxygen which dominates the broadening.

To use the new mechanism in practice, broadening and shift parameters for all broadening gases have to be provided by a line catalogue. We have compiled such a catalogue. Details of the compilation are presented in Section 2.2 below.

     It should be noted that both the classical and the revised broadening and shift calculation approach are available with ARTS 2.2 and will be kept in future versions. The approach applied in the actual calculation is governed by the format of the spectral line data provided (for further details see Section 2.2) and requires no specific settings by the user. Since data formats

for different line transitions are allowed to differ, it is possible to apply both line calculation approaches within one model run. Having both mechanisms available also simplified the testing of the new and more complex treatment, in order to ensure that the results are consistent with the old treatment where they should be.

## 2.2    Line catalogue

ARTS has its own internal representation of spectral line data that maps naturally to a native catalogue format. Two variants of

25 this internal catalogue data exist, corresponding to the two line broadening and shift algorithms introduced above.

     Beside other spectroscopic parameters, the catalogue format related to the classical algorithm, called ARTSCAT-3, contains the air broadening and shift parameters $\gamma_a$, $n_a$, and $\delta\nu$ representative for Earth conditions. Besides its internal formats, ARTS can digest other catalogues with different formats, e.g. the HITRAN format. These other databases typically report 'classical' Earth-representative spectroscopic parameters, hence their data are internally converted to ARTSCAT-3 format. A detailed

description of the ARTSCAT-3 format is given in Eriksson et al. (2011b).

     The ARTS internal catalogue format corresponding to the revised line broadening and shift algorithm, called ARTSCAT-4, reports broadening and shift parameter for individual foreign species. As already stated above, the currently covered broadening species are the most abundant species in the atmospheres of Venus, Earth, Mars, and Jupiter, namely $N_2$, $O_2$, $H_2O$, $CO_2$, $H_2$, and He. The complete format definition is given in Table 3.

**Table 3.** ARTSCAT-4 spectroscopic line data format. Column #0 gives the line entry start marker, the following parameters are separated by one or more blanks.

| Column | Parameter | Symbol | Unit |
|---:|---|---:|---:|
| 0 | '@' | - | - |
| 1 | molecule & isotopologue tag | - | - |
| 2 | center frequency | $\nu_0$ | Hz |
| 3 | line intensity | $S_0$ | $Hz\,m^2$ |
| 4 | reference temperature | $T_{ref}$ | K |
| 5 | lower state energy | $E_l$ | J |
| 6 | Einstein A-coefficient | $A$ | 1/s |
| 7 | Upper state stat. weight | $g_u$ | - |
| 8 | Lower state stat. weight | $g_l$ | - |
| 9 | broadening parameter self | $\gamma_s$ | Hz/Pa |
| 10 | broadening parameter $N_2$ | $\gamma_{N2}$ | Hz/Pa |
| 11 | broadening parameter $O_2$ | $\gamma_{O2}$ | Hz/Pa |
| 12 | broadening parameter $H_2O$ | $\gamma_{H2O}$ | Hz/Pa |
| 13 | broadening parameter $CO_2$ | $\gamma_{CO2}$ | Hz/Pa |
| 14 | broadening parameter $H_2$ | $\gamma_{H2}$ | Hz/Pa |
| 15 | broadening parameter He | $\gamma_{He}$ | Hz/Pa |
| 16 | broadening temp. exponent self | $n_s$ | - |
| 17 | broadening temp. exponent $N_2$ | $n_{N2}$ | - |
| 18 | broadening temp. exponent $O_2$ | $n_{O2}$ | - |
| 19 | broadening temp. exponent $H_2O$ | $n_{H2O}$ | - |
| 20 | broadening temp. exponent $CO_2$ | $n_{CO2}$ | - |
| 21 | broadening temp. exponent $H_2$ | $n_{H2}$ | - |
| 22 | broadening temp. exponent He | $n_{He}$ | - |
| 23 | frequency pressure shift $N_2$ | $\delta\nu_{N2}$ | Hz/Pa |
| 24 | frequency pressure shift $O_2$ | $\delta\nu_{O2}$ | Hz/Pa |
| 25 | frequency pressure shift $H_2O$ | $\delta\nu_{H2O}$ | Hz/Pa |
| 26 | frequency pressure shift $CO_2$ | $\delta\nu_{CO2}$ | Hz/Pa |
| 27 | frequency pressure shift $H_2$ | $\delta\nu_{H2}$ | Hz/Pa |
| 28 | frequency pressure shift He | $\delta\nu_{He}$ | Hz/Pa |
| 29 | quantum number information | - | - |

As part of the planetary toolbox, spectroscopic data have been compiled and made available with the arts-xml-data package. This is not the first effort to create a dedicated spectroscopic line list for ARTS: already in 2005, an ESA funded study lead to a

**Table 4.** Overview of the absorption species covered by the ARTS spectroscopic database. For 'planet interest' species, ARTSCAT-4 type data with foreign species specific line parameters has been compiled, while data for 'Earth-only' species has been taken from HITRAN without modifications and is provided in ARTSCAT-3 format. Empty data files are provided for 'no transition' species, which exhibit no absorption lines within the spectral region of interest of the planetary toolbox, but have to be considered as perturbing species. Species with 'ARTS 2.0' history are known species in ARTS' pre-toolbox version. 'New' species have been added in ARTS 2.2 with species data taken from HITRAN (default) or other sources like the JPL database (denoted by '*').

| species group | history | species |
|---|---|---|
| planet interest | ARTS 2.0 | $H_2O$, $CO_2$, $O_3$, CO, $CH_4$, $O_2$, $SO_2$, $NH_3$, HF, HCl, OCS, $H_2CO$, $H_2O_2$, $PH_3$, $H_2S$, $HO_2$, $H_2SO_4$ |
| | new | SO*, $C_3H_8$* |
| Earth-only | ARTS 2.0 | $N_2O$, NO, $NO_2$, $HNO_3$, OH, HBr, HI, ClO, HOCl, HCN, $CH_3Cl$, HCOOH, O, HOBr |
| | new | $CH_3OH$ |
| no THz transition | ARTS 2.0 | $N_2$ |
| | new | $H_2$, He* |

dedicated line list for millimeter/sub-millimeter limb sounding instruments (Perrin et al., 2005; Verdes et al., 2005). However, the old line list covered only selected bands, whereas the new line list covers a much broader spectral range.

In line with the scope of the planetary toolbox, to provide tools and data for propagation modeling in the atmospheres of Venus, Mars, and Jupiter as well as Earth in the spectral domain up to 3 THz, the line catalogue has been generated for gaseous absorption species considered of interest in these planets' atmospheres and for the range of atmospheric conditions of these planets. An overview of the species considered is given in Table 4.

The foreign species specific spectroscopic line parameters have been compiled from literature or extracted from the HITRAN (HIgh-resolution TRANsmission molecular absorption database, Rothman et al., 2009, 2013), GEISA (Gestion et Etude des Informations Spectroscopiques Atmosphériques, Jacquinet-Husson et al., 2011), and JPL (Jet Propulsion Laboratory, Pickett et al., 1998) spectroscopic databases. The sources of the data are given explicitly and in detail for each molecule in Mendrok and Eriksson (2014). Selected examples of the compilation procedure are detailed below. Species of obvious planetological interest but without line absorption signatures in the THz region, like ethane, germane, ethylene, or benzine, have been neglected.

In order to be able to also use the database for Earth applications, species only relevant in the Earth atmosphere, but none of the other planets (see Table 4) have been included as well. Line parameters of these species have been taken from the HITRAN edition current at the time of compilation (Rothman et al., 2013, update 13.06.2013) and converted to ARTSCAT-3 format without any further changes. Note that using ARTS functionality, users themselves can create ARTSCAT spectroscopic files from HITRAN data, e.g. from more recent editions or updates.

Foreign species specific line parameters have been derived by a careful literature investigation searching for experimental or theoretical studies specifically devoted to line broadening and shift by He, $H_2$, $CO_2$, or $H_2$. Furthermore, air broadening and

shift parameters reported in the HITRAN database are often deduced from individually determined and reported $N_2$ and $O_2$ broadening and shift data. In such cases, we applied the original $N_2$ and $O_2$ literature data in our catalogue compilation. For some combinations of gas species, absorption line and perturbing gas, the broadening and line shift parameters are absent in the literature, simply because spectroscopic studies dealing with these line parameters were never performed. In this case, the values quoted in our catalogue have been reasonably estimated, where the estimation strategy could differ from one absorption species to the other.

In particular, for the line broadening parameters, the values have been estimated from those existing in the literature for similar molecules or transitions. For water vapor, for example, numerous experimental and theoretical studies deal with its pressure broadening by $CO_2$ (Gamache et al., 2011, and references therein). Comparing the air and $CO_2$ broadening parameters we estimate them being related by $\gamma_{CO2} \sim 1.55\gamma_a$, and we used this expression in our catalogue compilation to derive $\gamma_{CO2}$ anytime it is otherwise unknown. Similarly, for water vapor transitions we estimated from the existing literature values $\gamma_{N2} \sim 1.1016\gamma_a$, $\gamma_{O2} \sim 0.6178\gamma_a$, and $\gamma_{He} \sim 0.24\gamma_a$. For ozone, $\gamma_{N2} \sim 1.029\gamma_a$ and $\gamma_{O2} \sim 0.89\gamma_a$ were deduced from the literature. We applied these relations to derive the respective $\gamma_i$ from $\gamma_a$ quoted in HITRAN for all water and ozone lines for which this information is otherwise missing. For ozone and other molecules, for which no $\gamma_{He}$ data exist in the literature, a default relation of $\gamma_{He} = 0.25\gamma_a$ was used to estimate $\gamma_{He}$. Regarding $\gamma_{N2}$ and $\gamma_{O2}$, we carefully checked that their values are consistent with their HITRAN $\gamma_a$ counterpart, i.e. to fulfill the condition $\gamma_a = 0.79\gamma_{N2} + 0.21\gamma_{O2}$ whenever this scaling strategy was applied.

For several linear molecules, like CO, HCl, HF and $CO_2$, a polynomial dependence of the $N_2$, $O_2$ and $CO_2$ broadening parameters on the rotational quantum numbers $m$ was established from measurements reported in the literature (Le Moal and Severin, 1986; Varanasi, 1975). For our compilation, we derived the $\gamma_{N2}$, $\gamma_{O2}$, and $\gamma_{CO2}$ using these expressions.

For some other molecules, e.g. $SO_2$, very precise line broadening parameters exist in the literature, however only for a very restricted set of rotational transitions when compared to the full list of lines in the spectral region up to 3 THz. Clearly, it is not possible to estimate the rotational dependence of these broadening parameters from these limited data. In these cases, the mean values deduced from the experimental data were implemented in our catalogue compilation.

For cases when the pressure broadening parameter of a perturber is not known at all, the default value adopted in HITRAN ($\gamma_i = 0.1\,\mathrm{cm}^{-1}$/atm corresponding to $\gamma_i = 30000\,\mathrm{Hz/Pa}$ in terms of SI units as applied in ARTS) was used. Similarly, the default value $n_i = 0.75$ was set for the pressure broadening temperature exponent. One exception here is helium, for which the default value was estimated as $\gamma_{He} = 0.04\,\mathrm{cm}^{-1}$/atm (12000 Hz/Pa). The pressure shift parameter $\delta\nu$, which is often unknown in the THz region for most of the perturbing gases considered here, has been set to a default value of zero in the absence of any data in the literature.

As indicated above, the primary source for perturber independent parameters like line positions and intensities has been the HITRAN and GEISA databases. However, several molecules considered of interest in planetary atmospheres are so far not covered by HITRAN or GEISA. This concerns for example sulfur monoxide (SO), sulfuric acid ($H_2SO_4$), propane ($C_3H_8$) and phosphine ($PH_3$). To generate the linelists for our catalogue, we used the line positions and intensities quoted in the JPL catalog. The line shape parameters were implemented using the same procedure as described above.

It should be noted that the applied strategy — preferring explicit per-species broadening and shift parameters over deriving them from HITRAN $\gamma_a$ as well as occassional application of parameterisations in terms of quantum numbers — can lead to differences in Earth atmospheric absorption cross sections when calculated from the toolbox catalogue compared to purely HITRAN-based calculations.

Along with the toolbox development, ARTS' list of known absorption species has been revised. It was updated with data from the recent HITRAN (Rothman et al., 2009, 2013) and TIPS (Total Internal Partition Sums, Fischer et al., 2003; Laraia et al., 2011) editions, which introduced a number of new species and isotopologues. Some further species not (yet) in HITRAN, but required for the planetary toolbox, have been added with species data (molecular mass, isotopologue ratio, partition function information) taken from the JPL spectroscopic database (Pickett et al., 1998, retrieved from http://spec.jpl.nasa.gov/) or from

educated guesses. The latter regards species that were rated as being of interest in the atmospheres of the toolbox planets and for which spectroscopic line data have been collected (e.g., $C_3H_8$), but also inert species that are required for the planet-suitable line broadening and shift algorithm introduced (e.g., He). Newly added species are identified in Table 4.

The spectroscopic catalogue data are available from the arts-xml-data package, where data are organised into one file per absorption species. It should be noted that our spectroscopic catalogue is a snapshot in time of the available spectroscopic data

of interest for planetary atmospheric remote sensing, at the time of development. The snapshot is as of early 2012, when the catalogue was compiled.

HITRAN, the most commonly used general spectroscopic line database has been undergoing very significant development in recent years (Hill et al., 2013). The new 2016 edition for the first time includes explicit broadening parameters for $H_2$, He, and $CO_2$ (Gordon et al., 2017), as well as many other new crucial parameters, for example for handling line mixing.

We enthusiastically welcome the new HITRAN paradigm, since it means that it will be possible to drive the new broadening calculation in ARTS with parameters directly from HITRAN in the future. The ARTS interface to the new HITRAN is not yet available, but will be worked on with high priority.

## 2.3  Refractivity

Changes in the propagation speed of electromagnetic radiation can lead to a bending of the propagation path, called refraction.

This is quantified by the refractive index $n = c/\nu_p$ or the refractivity $N = n - 1$, where $\nu_p$ and $c$ are the propagation speed in the medium and in vacuum, respectively. Neutral gases as well as free electrons contribute to refraction in planetary atmospheres.

Assuming that the refractivity of a gas is proportional to its density (e.g. Newell and Baird, 1965; Stratton, 1968), it can be determined from the refractivity at reference conditions (pressure $p_{ref}$ and temperature $T_{ref}$) and applying a gas law to scale it to other conditions. For a gas mixture, the total refractivity can then be determined as the sum of all partial refractivities, i.e.

$$N = N_{ref,1} \frac{n_1}{n_{ref,1}} + N_{ref,2} \frac{n_2}{n_{ref,2}} + \cdots, \tag{5}$$

where $N$ is the total refractivity, $N_{ref,i}$ is the partial refractivity for gas $i$ at reference conditions, $n_i$ is the partial density, and $n_{ref,i}$ is the reference density.

For Earth's atmosphere, commonly empirical parametrizations are applied that summarise the air, or at least its dry part, into one component scaled by the total pressure. Water vapor is often considered as a separate component due to its different reference refractivity and its strong variability in abundance (e.g., Thayer, 1974; Mathar, 2007). With $N_{\mathrm{ref},i}$ being specific to the gas species and varying notably between different species, it is obvious that further refined or generalized models are necessary when atmospheric composition is fundamentally different from Earth.

In ARTS 2.2, we have implemented the approach outlined in Equation 5 with species $i$ being individual atmospheric gas species. The effect of this for the refractivity profile of different planets is shown in Figure 3.

Reference refractivities of $N_2$, $O_2$, $CO_2$, $H_2$, and He, derived at 47.7 GHz and considered to be valid for microwave and submillimeter-wave frequencies, have been taken from Newell and Baird (1965). To achieve a better agreement with parametrisations for Earth, $H_2O$ is considered, too, and its reference refractivity has been estimated from the parametrisation by Thayer (1974). It is $H_2O$ that is causing the kink at high densities (near the surface) for Earth in Figure 3.

For scaling to non-reference conditions, we apply the ideal gas law yielding

$$N = \frac{T_{\mathrm{ref}}}{p_{\mathrm{ref}}} \sum_{i=1}^{m} N_{\mathrm{ref},i} \frac{p_i}{T} \; , \tag{6}$$

where $p_i$ is the actual partial pressure of species $i$ and $T$ the actual temperature. To account for missing contributions of unconsidered species, the refractivity derived from Equation 6 is normalized to a total volume mixing ratio of 1, similar to the line broadening normalisation in absorption calculations (see Section 2.1). ARTS offers further models specifically for Earth air refractive indices for the microwave (Thayer, 1974) and the infrared spectral region.

Electron contributions are negligible for passive observation techniques, but play a recognizable role for some active techniques like radio links and Global Navigation Satellite System (GNSS) measurements (discussed in Section 3.1).

Neglecting influences of any magnetic field, the refractive index of a plasma like the ionosphere is (e.g., Rybicki and Lightman, 1979)

$$n = \sqrt{1 - \frac{N_e e^2}{\epsilon_0 m \omega^2}} \; , \tag{7}$$

where $\omega$ is the angular frequency ($\omega = 2\pi\nu$), $N_e$ the electron density, $e$ and $m$ the charge and the mass of an electron, respectively, and $\epsilon_0$ the permittivity of free space. This refractive index, which is less than unity but approaching unity with increasing frequency, describes the phase velocity of the radiation, hence determines the ray path.

The propagation speed of the signal energy through the plasma, which determines signal delays along the path, is described by the group velocity and the corresponding group refractive index (Rybicki and Lightman, 1979)

$$n_{\mathrm{g}} = \left(1 - \frac{N_e e^2}{\epsilon_0 m \omega^2}\right)^{-1/2} = \frac{1}{n} \; . \tag{8}$$

The electron contributions to the phase and the group velocity index of refraction according to Equations 7 and 8 have been implemented in ARTS 2.2.

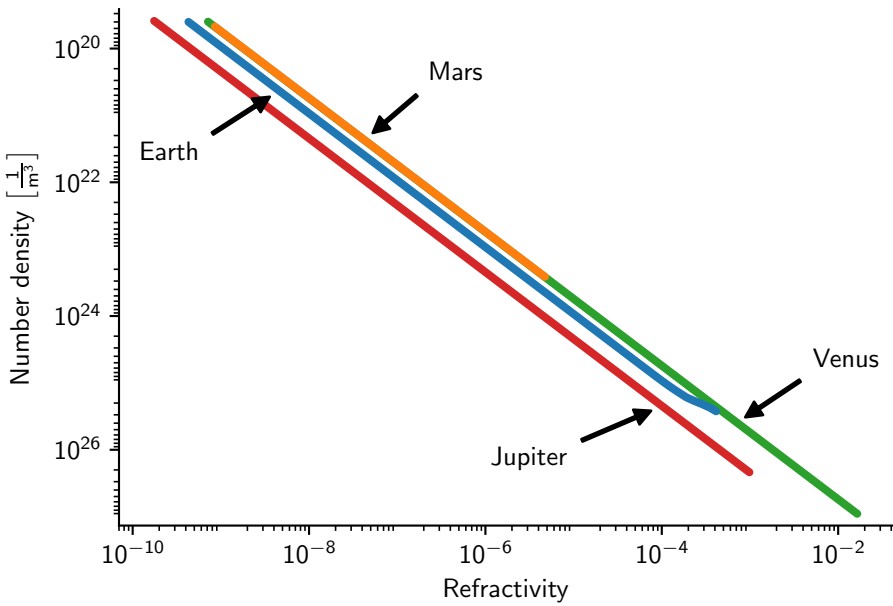

**Figure 3.** Refractivity profiles for different planets. Density is used here for the vertical scale instead of pressure, because otherwise the figure is complicated by the widely different temperatures on different planets leading to different densities at similar pressures. The model atmospheres here are the same as in Figure 1 and described in the caption of Table 2. Data begin at the planets surface, except for Jupiter where they begin at the lowest level of our model atmosphere.

## 2.4 Isotopologue abundances

Absorption coefficients are proportional to the amount of the absorption species and the transition line strength. For practical reasons, the amount is often provided in terms of the volume mixing ratio (VMR) of the species covering all isotopologues, e.g. water vapor VMR instead of the VMR of specific isotopologues like $H_2{}^{16}O$ or HDO. Then, scaling by the relative abundance of the isotopologue is required. The scaling can either be applied to the VMR or the line strength. HITRAN implements the latter approach by providing pre-scaled line strenghts valid for mean Earth conditions. However, as isotopologue abundances differ between planets, this approach is inflexible and inconvenient for planetary applications. ARTS on the other hand applies the VMR scaling approach requiring isotopologue abundance independent line strengths.

For being able to apply HITRAN spectroscopic data, ARTS contains a hard-coded table of relative isotopologue abundances in the Earth atmosphere, where the relative abundance is the ratio of abundance of an isotopologue to the abundance of the gas species over all its isotopologues (in contrast, isotopologue *ratio* refers to the abundance ratio of an isotopologue to the abundance of the main isotopologue of the species). This table is used to convert HITRAN isotopologue scaled line strenghts into ARTS' abundance independent line strengths. In previous ARTS versions, the table was also applied in the VMR scaling of absorption coefficients. In ARTS 2.2, isotopologue abundance has been introduced as a user-accessible variable, which can

**Table 5.** Planetary isotopic ratios as applied in the isotopologue abundance data table generation. Values for Venus, Earth and Mars taken from Lammer et al. (2008, Table 1), Jupiter from Owen and Encrenaz (2003). The Earth values are given for reference only, because in the actual table generation we inferred them from the HITRAN Earth Isotopologue abundance for each individual molecule.

| Planet | D/H | $^{15}N/^{14}N$ |
|--------|-----|-----------------|
| Venus | 1.9e-2 | as Earth |
| Earth | (1.5e-4) | (3.7e-3) |
| Mars | 8.1e-4 | 5.7e-3 |
| Jupiter | 2.6e-5 | 2.25e-3 |

be initialized from the built-in isotopologue table, e.g. for Earth atmosphere calculations, or read from file, e.g. for planetary use.

As part of the planetary toolbox, tables of relative isotopologue abundances for Venus, Mars, and Jupiter are provided with the arts-xml-data package. We generated these, based on the available planetary literature. What can readily be found there are not isotopologue abundances for all different molecules, but rather isotope ratios for important atoms, for example the ratio of deuterium to normal hydrogen. From these, we generated the molecular isotopologue tables by rescaling Earth isotopologue abundances with the planetary isotope ratios reported in Table 5.

Isotope ratios of D (in all planets) and $^{15}N$ (in Mars and Jupiter) were found to significantly differ from Earth values, while other species are within 5% of their Earth values. Adaptation of isotopologue abundances was, hence, restricted to species containing hydrogen and nitrogen.

For spectral lines belonging to molecules that contain heavy hydrogen or nitrogen atoms, the change in absorption due to these abundance differences can be very significant. To give an example, the isotopologue abundance of HDO is more than 100 times that of Earth on Venus, 5 times that of Earth on Mars, and only less than 0.2 that of Earth on Jupiter. Because absorption is proportional to abundance, these differences translate directly into absorption differences for spectral lines belonging to this species.

## 2.5 Further adaptations for planetary use

Several other planet dependent parameters have also been turned into user-controlable parameters. This includes size and shape parameters of ellipsoidal planets, required for line of sight calculations, where also predefined settings for the toolbox planets in the form of dedicated workspace methods are available. This furthermore concerns settings of the gravitational constant and of the molar mass of dry air, both required for deriving altitude-pressure relations assuming hydrostatic equilibrium, as well as the sideareal rotation period of a planet, required for considering Doppler shifts resulting from the rotation of a planet observed from a platform not in orbit around this planet (see Section 3.2.4).

# 3 Further new model features and remaining restrictions

Besides the adaptations described above, which were necessary to make the propagation model applicable to general planetary atmospheres, several other new modeling features are available with the new ARTS version. An example is the addition of radio occultation measurements and radio link budget estimations (Section 3.1), which is of particular interest for the planetary toolbox since such measurements are relevant for planetary exploration (e.g. Eshleman et al., 1987; Hinson et al., 1997; Oschlisniok et al., 2012). Some physical processes affect both passive and active measurements. The fact that ARTS uses identical algorithms to model these processes provides consistent simulations of both techniques.

In this release, some physical processes that were not treated before have been added. These include for example Doppler shifts due to wind and planet rotation, the oxygen Zeeman effect, Faraday rotation, and dispersion, which all are described in Section 3.2. In order to model several of these effects, additional model input characterising the atmosphere is required. Section 3.3 provides details on the handling of these input parameters.

Active measurement techniques provide more diverse measurement parameters. Therefore, the measurement module output has been extended. This also allows for more detailed output for passive measurement simulations. An overview is given in Section 3.4.

## 3.1 Radio link budgets

A basic handling of radio link budgets has been introduced. The implementation focuses on the attenuation of the power between a transmitter of a coherent signal and the receiver position, but also some other aspects are covered by the auxiliary variables provided. The latter includes a basic treatment of radio occultation, i.e. when a coherent microwave signal, such as from GNSS, is recorded by either a satellite- or ground-based receiver in order to determine certain atmospheric properties (e.g. Kursinski et al., 2000; Nilsson and Elgered, 2008). Only an overview of these additions is given here, for details see Eriksson et al. (2011c) and the built-in documentation.

The most critical step of these calculations is to establish the propagation path between transmitter and receiver. This step is so far only handled by a quite simple and time consuming algorithm (Eriksson et al., 2011c) and only considers effects covered by geometrical optics. Snell's law is used to determine the bending of the radiation as it travels through the atmosphere, and the algorithm looks for a path that connects transmitter and receiver. In reality, there can be more than one possible path for atmospheres with strong vertical temperature gradients (so called multi-pathing), but this is currently not treated in ARTS. The algorithm simply finds a link path, or determines that no path is possible, due to interception by the planet's surface.

The receiver and transmitter can be placed at arbitrary positions, allowing that, for example, satellite-to-satellite as well as aircraft-to-ground radio links can be analysed. All atmospheric dimensionalities are handled (1D, 2D and 3D).

Attenuation due to gases and particles are included exactly in the same manner as for pure transmission calculations, but an important additional attenuation term, that is in fact dominating, is the 'free space loss'. This term is in ARTS defined as $1/(4\pi l^2)$, where $l$ is the distance along the line of sight. A probably more common definition of the term, based on the 'Friis transmission formula', is $(\lambda/(4\pi l))^2$, where $\lambda$ is the wavelength (see e.g. Ulaby et al., 2014, Sec. 3.3). We avoid the later

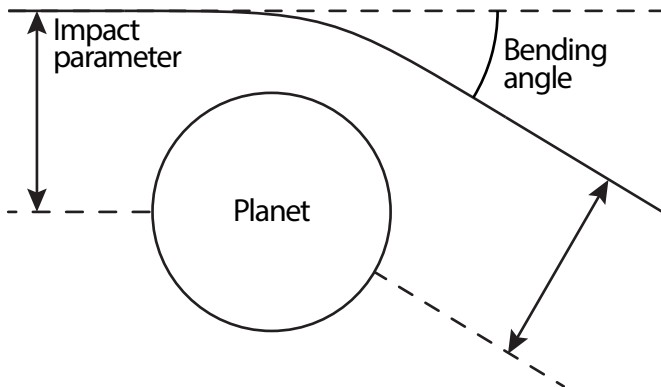

**Figure 4.** The radio occultation geometry with impact parameter and bending angle. See for example Kursinski et al. (2000) for details.

version because the additional $\lambda/(4\pi)$ forefactor is meant to account for the variation of the receiver's gain with frequency, while in ARTS the ambition is to keep atmospheric and sensor effects strictly apart.

A special effect when transmitting coherent signals is (de)focusing. Simply expressed, the effect originates in the fact that refraction can vary over the wavefront. Defocusing occurs if neighbouring ray-paths in a medium diverge more quickly than for
free space propagation. The opposite, focusing, can also take place, but is in general less pronounced. ARTS provides a general and rough estimate of (de)focusing by simply determining the propagation path at two slightly shifted propagation angles, starting at the transmitter, and comparing the distance between the two paths, at the receiver, to the distance expected from pure geometry. For satellite-to-satellite links, the user can instead select to make use of some standard analytical approximations (e.g. given in Kursinski et al., 2000, Sec. 3.7), where both the defocusing and focusing components are considered.

For a more complete characterisation of the radio link, the auxiliary output at hand includes the following quantities: bending angle, impact parameter (both defined as in Figure 4), extra path delay, Faraday rotation (see Section 3.2.2) and all loss terms reported individually.

An application example of this is shown in Figure 5, which makes use of operational atmospheric analysis data from the European Centre for Medium-Range Weather Forecasts (ECMWF), as well as a co-located radio-occultation observation from
the GNSS Receiver for Atmospheric Sounding (GRAS). In the left panel, it shows the ARTS-simulated bending angle, based on the ECMWF model data, together with observed bending angles for two different GRAS data retrieval algorithms (so-called geometric optics (GO) and full spectrum inversion (FSI)).

In the right panel, Figure 5 shows the observed transmitted power from the GRAS instrument, as well as the ECMWF-model-based ARTS simulation, broken down by individual effect. Free space loss is the dominating attenuation mechanism, but varies
little during a GRAS occultation. The 'power' in Figure 5 is normalised to the free space loss at a high altitude. Atmospheric attenuation (absorption of gases, no scattering included in these calculations) is low in the stratosphere, but is an important factor for low impact heights. However, the main variation of power during the occultation is determined by defocusing.

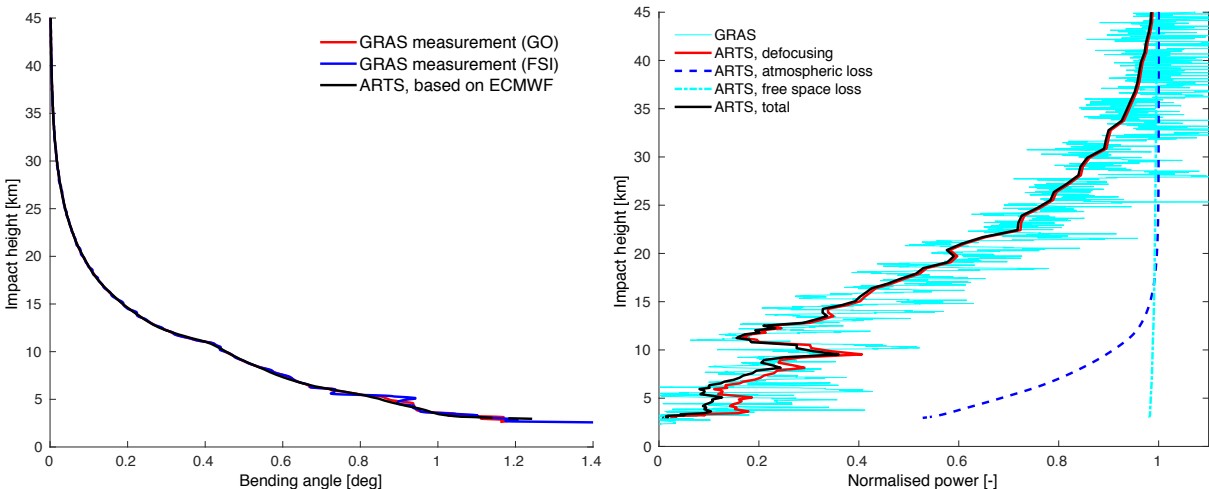

**Figure 5.** Comparison of ARTS forward model calculations with GRAS data. The input to ARTS is ECMWF operational analysis data. The left panel compares ARTS simulations to observed bending angles determined in two different ways from raw GRAS observations. The right panel shows the observed and simulated link budget for transmitted power. As common for radio-occultation observations, the vertical scale, impact height, is simply the impact parameter of Figure 4 minus the Earth radius.

The actual GRAS measurement is also affected by scintillations (explaining the most rapid variations of the power), but this mechanism is not treated by ARTS.

## 3.2 General radiative transfer features

This section presents new features in ARTS that are of radiative transfer character and of general applicability, i.e. that can be used together with the different measurement technique modules.

### 3.2.1 Handling of non-particle polarisation

Older ARTS versions have assumed that only particulate matter (including the planet's surface) causes effects that go beyond a scalar description. That is, gaseous absorption has been seen as scalar attenuation coefficients. This limitation is now removed, for two reasons. First of all, the Zeeman effect (Section 3.2.3) causes the absorption to depend on polarisation, which cannot be described in a scalar manner. Secondly, effects of magneto-optical (del Toro Iniesta, 2003, Sec. 1) character also do not fit into a scalar formalism, and both Faraday rotation and parts of the Zeeman effects fall into this category.

Accordingly, the code has been revised to throughout allow for a matrix description of propagation effects. As a consequence, the terminology used in ARTS has also changed, what was before denoted as 'absorption coefficient' is now called 'propagation matrix' (following e.g. del Toro Iniesta, 2003) to reflect the wider scope of the associated variables and methods. This extension made it also possible to add a feature treating particulate matter as purely absorbing matter. This allows for a much faster treatment of radiative transfer when scattering is neglected. This simplification is only valid when the particles are small

compared to the wavelength, i.e. their single scattering albedo is small. There is no check in ARTS, though, that this condition is fulfilled; this judgement is fully left to the user.

### 3.2.2 Faraday rotation

A wave propagating through the ionosphere will force free electrons to move in curved paths. For example, if the incident wave is circularly polarised, the motion of the electrons will be circular. As a consequence, the refractive index is not a single constant, but depends on polarisation. A manifestation of this 'Faraday effect' is that the electric field vector of a wave propagating through the ionosphere will rotate. This is denoted as Faraday rotation, and this physical mechanism is now handled by ARTS. The core expression is (e.g. Rybicki and Lightman, 1979)

$$r = \frac{e^3}{8\pi^2 c\epsilon_0 m^2 f^2} n_e(s)\mathbf{B}(s)\cdot\hat{\mathbf{s}} \,, \tag{9}$$

where $r$ is the change in rotation angle [rad/m], $e$ is the charge of an electron, $c$ is the vacuum speed of light, $\epsilon_0$ is the permittivity of vacuum, $m$ is the electron mass, $f$ is the frequency, $n_e(s)$ is the density of electrons at point $s$, $\mathbf{B}$ is the magnetic field, $\cdot$ denotes the dot (scalar) product and $\hat{\mathbf{s}}$ is the unit vector along the propagation direction. That is, the rate of rotation depends on the number of free electrons and the angle between propagation and magnetic field directions. It is also propertional to $f^{-2}$, causing Faraday rotation to be negligible above about 3 GHz (Ulaby et al., 2014). See Eriksson et al. (2011c) for further details.

### 3.2.3 Zeeman effect

Molecules with unpaired electrons experience an effect named the Zeeman effect after its discoverer (Zeeman, 1897). The Zeeman effect polarises the radiation as a function of magnetic field orientation, and splits what is otherwise a single spectral line into several lines, with a splitting distance that is a function of the magnetic field strength. The total line strength is kept constant but distributed over the split lines. From an ARTS user perspective, the main practical requirement for Zeeman calculations is that the magnetic field must be specified as an additional input field. Some additional spectroscopic parameters are also needed.

The physical mechanism, from which the effect arises, is that the spin of the unpaired electrons couples to the external magnetic field, changing the energy state of the molecule as a function of magnetic field strength by

$$\Delta E = -gM|\mathbf{B}|\mu_b \,, \tag{10}$$

where $g$ is the state-dependent Landé factor, $M$ is the projection of the total angular momentum number $J$ along the magnetic field, $|\mathbf{B}|$ is the magnetic field magnitude, and $\mu_b$ is the Bohr magneton. See e.g. Figures 4 and 7 in Larsson et al. (2014) for an example of how the Zeeman effect influences the brightness temperature signal as perceived by a sensor. $M$ belongs to the set $\{-J, -J+1, \cdots, J-1, J\}$, and can only change by $-1$, $0$, or $1$ during a transition. This makes for a total of $3(2J+1)$ lines in place of the single original line.

The change in projection of $J$ is related to the polarisation of the radiation and is influenced by the angle between the magnetic field and the path of propagation of the radiation. If the magnetic field is in the plane of observation, transitions with

a changing $M$ affect linear polarisation along the magnetic field and transitions with a constant $M$ affect linear polarisation perpendicular to the magnetic field. If the magnetic field is pointing directly towards or away from the observer, only transitions with a changing $M$ affect the radiation. This radiation will have its circular polarisation state altered. The implementation and the physics of the Zeeman effect in ARTS are described in detail by Larsson et al. (2014) and Larsson (2014), and references therein.

For this ARTS version, the only tested Zeeman absorption species is molecular oxygen ($O_2$). Other species, like NO and SO, are also Zeeman-affected (see, e.g. Veseth, 1977; Christensen and Veseth, 1978), and while ARTS should be able to model the effect also for these species, this is left to future versions.

The ARTS oxygen Zeeman calculations have been validated in some studies so far: Navas-Guzmán et al. (2015) simulated ground-based observations of mesospheric molecular oxygen spectra in linear polarization for several observational directions, and found good agreement with observations; Larsson et al. (2016) compared the ARTS simulations for a down-looking meteorological sensor to observations and to another, stronger parameterised Zeeman model. The module has also been applied to theoretical studies on mapping Martian surface magnetism (Larsson et al., 2013, 2017).

### 3.2.4 Doppler shifts

A basic treatment of Doppler shifts due to winds has existed in ARTS for some time. For ARTS 2.2 this part was completely recoded, and the Jacobian of observations with respect to the three standard wind components (u, v and w) can now also be calculated. The immediate motivation for this extension was the wind retrievals presented in Rüfenacht et al. (2014). The Doppler shift $\Delta\nu$ is given as

$$\Delta\nu = \frac{-v\nu_0\cos(\gamma)}{c} \,, \tag{11}$$

where $v$ is the wind speed, $\nu_0$ is the rest frequency and $\gamma$ is the angle between the wind direction and the line-of-sight. More details are found in Eriksson et al. (2011c). Note that the Doppler shift caused by the random thermal motion of air molecules is part of the line shape, the function describing the frequency dependence of the absorption of each transition, and is therefore not modeled explicitly here.

The rotation of the planet is another possible cause of Doppler shifts. This effect can be a concern for satellite measurements, but there is no net impact if the observer follows the planet's rotation, such as for ground-based observations of the planet's own atmosphere. In ARTS, this Doppler effect can be included by mapping the planet's rotation to a zonal wind speed, the u component. This pseudo-wind, $v'_u$, is calculated as

$$v'_u = \frac{2\pi\cos(\alpha)(r+z)}{t_p} \,, \tag{12}$$

where $\alpha$ is the latitude, $r$ is the local planet radius, $z$ is the altitude and $t_p$ is the planet's rotational period. This term is added to the true zonal wind speed.

Further, for moving observation platforms, such as aircraft or satellites, the sensor velocity can result in a significant Doppler shift and ARTS provides now a rudimentary handling of this aspect. However, the platforms are normally moving with a constant speed and the associated Doppler shift is probably most easily handled outside of the forward model.

### 3.2.5 Dispersion

By default, ARTS assumes that the propagation path is common for all frequencies, that there is no dispersion. In most cases this is a good approximation. The atmosphere is dispersive at frequencies around strong transitions, but as discussed in Buehler et al. (2005a), this effect can in practice be neglected, because it is associated with very high absorption.

5  However, the introduction of ionospheric refraction (Equation 7), which is frequency dependent, now demanded to add a feature to handle dispersion. This was solved by making it possible to optionally have frequency as the outermost loop in the calculation, so that propagation paths are recalculated for each individual frequency.

This solution is completely general, so that ionospheric dispersion can be combined with all other features of ARTS. It also means that dispersion can now be modeled explicitly even if ionospheric refraction is not included, if one is willing to pay the 10 price of significantly increased computational cost for the calculation of the individual frequency-dependent propagation paths.

### 3.2.6 The $n^2$ radiance law

ARTS has been corrected regarding passive observations, where the $n^2$ radiance law was not fully considered before. This law says that, even in the absence of attenuation, the radiance would change along the propagation path due to refraction effects. The preserved quantity is (Mobley, 1994; Mätzler and Melsheimer, 2006)

$$15 \quad \frac{I'}{n^2} \, , \tag{13}$$

where $I'$ is the uncorrected radiance and $n$ is the refractive index as defined in Section 2.3.

It can be shown that it suffices to consider the refractive index at the point of emission and the point where the measurement is performed (Mobley, 1994, Eq. 4.23). To incorporate the $n^2$-law in the description of emission turns out to be equivalent to replacing the local propagation speed with the speed of light in vacuum in the Planck blackbody expression. This feature was 20 already in place, but now also a scaling with $n^{-2}$ at the measurement position is applied in ARTS. This change affects only observations performed within the model atmosphere, as the relevant $n$ for satellite-based observations is unity. This particular treatment of the $n^2$-law is discussed further in Eriksson et al. (2011b).

### 3.2.7 Continuum models

A number of gas absorption continuum models are available with ARTS. This particularly covers models for the micro- and 25 millimeter-wave region (e.g. Liebe et al., 1993; Rosenkranz, 1993, 1998), but also several editions of the Clough-Kneizys-Davies continuum model (CKD, Clough et al., 1989, 2005), later enhanced by Mlawer and Tobin (MT_CKD, Mlawer et al., 2012), models that cover the entire millimeter to infrared spectral range. However, all of these continua have been developed with focus on Earth observations. In atmospheres with different major atmospheric constituents as well as pressure and temperature conditions, different continua play important roles.

30  Recent editions of the HITRAN database offer collision-induced absorption (CIA) data (Richard et al., 2012). CIA is caused by collisions of centro-symmetric molecules that possess no permanent electric dipole, like $O_2$, $N_2$, $H_2$, $CO_2$, and $CH_4$, but for

which collisions create a transient dipole. The absorption strength of CIA is characterised by its dependence on the molecular density of both molecular species involved in the collision:

$$\alpha_{i,j} = \kappa_{i,j}\, n_i\, n_j \,,$$ (14)

where $i$ and $j$ denote the two species involved, $\alpha$ is the absorption coefficient, $\kappa$ the binary absorption cross section, and $n$ the number density of the respective species.

Tabulated frequency and temperature dependent $\kappa_{i,j}$ for a variety of species $i$ and $j$ are available from recent HITRAN editions. For the planetary toolbox with a focus on Venus, Mars and Jupiter, CIA data for $CO_2$-$CO_2$, $H_2$-$H_2$, and $H_2$-He are of particular interest. A mechanism to consider bi-species dependent absorption has been implemented, where tabulated binary cross sections have to be provided to ARTS. A method to derive $\kappa_{i,j}$ from HITRAN data is available. To make the HITRAN data seamlessly work with ARTS, we created a slightly modified version of the data in native ARTS format: data sets covering the same frequency range have been merged into one frequency-temperature table, data sets only covering visible and shorter wavelengths have been removed, and all binary cross sections have been converted from HITRAN ($cm^5$/molec$^2$) to ARTS ($m^5$/molec$^2$) units. These data are available as part of the arts-xml-data package.

## 3.3  Extended atmospheric state characterisation

Several of the new physical processes described above require additional input parameters in order to be properly modeled. Doppler shifts from wind require characterisation of the wind, Zeeman effect and Faraday rotation require magnetic field knowledge, and Faraday rotation and ionospheric refraction require a description of the electron density.

All non-scattering atmospheric matter in ARTS is subsumed as absorption species with associated atmospheric fields gathered into a variable named vmr_field. Following this approach, free electrons have been added to the list of allowed absorption species, and when considering free electrons in a radiative transfer calculation, the electron density field ($m^{-3}$) is held as one entry in vmr_field.

Winds as well as the magnetic field are vector parameters, hence require three pieces of information per atmospheric grid point. For both parameters, variables have been created to hold the (up to) three dimensional fields of the individual vector components u, v, and w. Generally, parameters that are not explicitly set are interpreted as equivalent to zero winds and magnetic field components, respectively.

## 3.4  Auxiliary output

The main output of ARTS' radiative transfer part is a vector with all simulated observations appended, i.e. directly matching the 'measurement vector' **y** in the formalism of Rodgers (2000). Auxiliary data that is input or output of ARTS workspace methods, such as the observational position(s) associated to a radiative transfer calculation, are always at hand. However, in many cases there exists also an interest in additional information calculated inside the radiative transfer methods. A typical example is the optical thickness related to an observational setup. Other commonly requested quantities include absorption, temperature and volume mixing ratios along the propagation path.

A general approach for obtaining such additional auxiliary data has been added to ARTS. The exact set of auxiliary variables that can be obtained differs depending on what radiative transfer problem has been solved. For example, there is no need to cover optical thickness by methods that have atmospheric transmission as main output. The auxiliary variables at hand can be divided into two main classes. The first class is the variables defined along the propagation path (such as temperature and partial transmissions). This class can just be obtained for single pencil beam calculations. This is because there is no common propagation path in the general case, considering a finite field of view, where a weighting of results from different propagation paths with the antenna pattern is performed. The second class consists of quantities resulting in a scalar value for each simulated observation value, that can be provided also for simulations including weighting with sensor characteristics. This class includes optical thickness and flags reporting intersection with the ground or the 'cloud box'.

## 3.5 Remaining restrictions and outlook

Although ARTS is a fairly general radiative transfer model, several important restrictions remain. Perhaps the biggest one is that there is no collimated beam radiation source, so the model is not suitable for modelling the scattering of solar radiation in the atmosphere. Version 2.2, the subject of this article, also does not allow handling absorption and emission for conditions outside of local thermodynamic equilibrium (non-LTE). This may change in a future release, because work in this direction is ongoing.

Line mixing, a phenomenon where closely spaced molecular energy levels affect the shape of the spectral lines, is also not treated in this version. The phenomenon is important, because it affects two molecules of high scientific interest, $CO_2$ and $O_2$. For the former, line mixing is important for calculating correct energy fluxes, as needed by atmospheric circulation models and radiative-convective equilibrium models. For the latter, line mixing is important for temperature remote sensing in the microwave spectral range. This aspect is in active development, and future releases of ARTS are planned to include line mixing for both species.

Other less prominent restrictions are that ARTS does not handle birefringence, handles ionospheric propagation only at frequencies well above the plasma frequency, and does propagation path calculations only by geometric optics (it is not a wave optics propagator). Also, while the model has simple surface models (specular and Lambertian) or accepts general bidirectional surface reflectivity as input, there is no explicit subsurface model, which might be of interest for complex surfaces such as snow.

## 4 Summary and conclusions

This article describes version 2.2 of the atmospheric radiative transfer simulator ARTS. Its most significant innovation is the planetary toolbox, which allows radiative transfer simulations for other solar system planets, in addition to Earth, but should also benefit the studies of bodies beyond the solar system like exoplanets. The necessary adaptations have made the program more general in several important aspects, which benefits also applications for Earth's atmosphere.

Besides this extension, there were numerous improvements and developments compared to the last version that is described in the literature (Eriksson et al., 2011a), and the most important of these are also described.

We hope that others may find the model useful, and always appreciate comments, suggestions, and usage examples. The best way to get in touch with the model developers is via the ARTS website, given below.

*Code and data availability.* The model, together with extensive documentation, associated tools, and input data, is freely available under a GNU public license from www.radiativetransfer.org. The web page also hosts dedicated email lists for ARTS users and developers. This
article is about ARTS version 2.2, the exact subversion number at the time of writing is 2.2.64. There may be a limited number of bugfix releases that increment the last digit number, but the active development of new features is happening in version 2.3.

*Competing interests.* The authors declare that they have no competing interests.

*Acknowledgements.* We would like to acknowledge the tremendous support from the ARTS developer and user community. For example, they contribute by answering questions by new users on the ARTS email lists.
We also acknowledge financial support, most importantly from the European Space Agency (ESA) in the context of contract No 4000104175/11/NL/AF 'Microwave propagation toolbox for planetary atmospheres'. As part of this contract, Bengt Rydberg performed work that helped to implement handling of radio links. Stefan Buehler's contribution was also supported through the Cluster of Excellence CliSAP 328 (EXC177), Universität Hamburg, funded through the German Science Foundation (DFG).

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
