# Peer review of "ARTS, the atmospheric radiative transfer simulator — version 2.2, the planetary toolbox edition"

_Geoscientific Model Development, 2017_

## Short Comment (SC1) · 23 Oct 2017

Stefan

As explained in https://www.geoscientific-model-development.net/about/manuscript_types.html it is preferred for the code to be uploaded as a supplement or to be made available at a data repository with an associated DOI (digital object identifier) for the exact model version described in the paper. Given the impermanence of websites, it is not encouraged that authors put models on their own website. Authors should consider improving the availability with a more permanent arrangement.

[Figure]

On the website http://www.radiativetransfer.org/misc/download/stable/2.2/ several versions/releases of the code are available for download. The authors need to explicitly state in the manuscript which version the manuscript is based on and are encouraged to provide the code (possibly as stripped version of the tar-ball) as supplement.

Lutz Gross GMD Executive Editor

---

## Referee Comment (RC1) · Anonymous Referee #1 · 25 Oct 2017

Summary:

The article describes the current version of the ARTS package, which is a widely used line-by-line radiative transfer model for the thermal spectral range. The paper is not a complete model description but focuses on new features that have been included after the last ARTS reference publication in 2011. The major extension is a planetary toolbox which enables simulations for the planets Venus, Mars and Jupiter in addition to Earth.

The paper is interesting to read, and, as also mentioned in the abstract, it focuses on the major extension, the planetary toolbox, in particular, on how the line-by-line

calculations need to be adapted for other planets. My major concern is, that the reader, a potential ARTS user, could think, that this is the main application of ARTS. The introduction summarizes the history of the ARTS development, but afterwards I miss an overview of the features included in ARTS, version 2.2. I think that for example an overview table including the most important ARTS features (not only the new ones) should be added. Further it would be nice to include examples of use of the new features, if possible including example input files as supplementary data. The first section should also include something like "basic usage", how the program is installed, how it is executed, etc ...

I have several more specific comments and recommend to publish the paper after the revisions.

General Comments:

Regarding the planetary toolbox extension, I have general questions: I understood that in detail the absorption lines look different for planetary atmospheres other than Earth because one has to take into account different line broadening parameters and also isotopic ratios are different. It is not clear from the paper how large the resulting differences in the absorption lines actually are.

A further question is, whether it is possible to estimate the error, which would be obtained for broad-band simulations, when the radiance is integrated over a large number of spectral lines. In particular IR instruments often measure broader bands. Does the error for integrated radiance become larger or smaller than for individual lines? Can ARTS without adaptions for other planets be used to simulate IR broad-band observations of other planets?

Specific comments:

- Abstract: Please motivate, why you have included Faraday rotation and Zeeman splitting. Why are these features needed for atmospheric radiative transfer?

[Figure]

p.1., l.16: "computer codes for energy flux computations and remote sensing simulations have been developed in parallel..." - Is this really true? Traditional solution methods like the discrete ordinate method (theory by Chandrasekhar 1950, included e.g. in DISORT code) compute radiance and irradiance simultaneously. I guess that there are more codes that calculate both, e.g. also libRadtran gives radiances, and can also compute (spectrally integrated) irradiances.

p.5, Eq. 2: Why has this equation been chosen to calculate the pressure shift in ARTS. Where do the constants 0.25 and 1.5 come from?

Sec. 2.1: Could you include an example plot showing the difference between the two approaches to calculate line broadening and line shift for a spectral line in the Earth atmosphere? How significant is the difference? Here it would also be nice to include ARTS input files (as supplement), showing how to set up the classical absorption calculation and what is changed for the new approach.

p7, l14ff: The line catalogue generated for planetary atmospheres includes data up to 3THz. Are there plans to extend it to the full IR region and possibly also to visible spectral region?

p.7 l.21: The sources of data for the planetary database are given in Mendrok et al. 2014. The reference is a technical report for ESA, thus it might be difficult to access it. Please include the data sources in the paper (a table would be nice, perhaps references could be included in Table 2).

p.11, Sec. 2.3, Refractivity: A plot showing refractive index profiles of all planets in the planetary toolbox would be interesting.

p.11, l. 26. Refer to section 3.1 so that reader knows that radio link simulations are actually possible with ARTS and thus the electron contribution to the refractivity is important to include in the model.

Sec. 2.4: Can you show an example (e.g. a specific water vapor line) and demonstrate

how this line changes with different isotopic ratios? I.e. plot the same line for Venus, Earth, Mars and Jupiter.

Sec. 2.5: "Doppler shifts" are mentioned here but at this point the reader does not know about this feature in ARTS. Please refer to Sec. 3.3.4.

Sec. 3.1: Please provide one example for a radio link budget simulation.

p. 14, l. 10: "considers effects covered by geometrical optics ..." - which effects, please describe more precisely. Do you mean Snell's law of refraction or something else? What is meant with "multi-pathing", please explain briefly.

p. 14, l.15/16: Is it correct that in one of the equations for the "free space loss" the denominator includes 4*pi and in the other (4*pi)^2?

p. 14, l.29: Please explain in a few words what is "Faraday rotation" and why it has to be considered for radio link budget simulations.

Sec. 3.2: Please provide one example for a radar simulation.

p. 15, l.9: Please give an example when $T_h$ and $T_a$ are not equal. How relevant is this inequality for radar simulations?

Sec. 3.1: Please provide an example for non-particle polarization. It would be very interesting to show the circular polarisation by Zeeman splitting. How large is the degree of circular polarisation? For which remote sensing applications is this important?

Sec. 3.3.6: I do not understand this section: Absence of attenuation means vacuum, right? In vacuum there is no refraction. How can the radiance change in absence of attenuation? Please explain.

p. 19, l.9: The "vacuum speed" is the "speed of light", right?

Sec.4. The summary and conclusions section is very short. It should include once more a general overview of what is ARTS, including LBL-absorption calculations, RT in

thermal range (microwave, IR), scattering by cloud droplets and (oriented) ice crystals, surface reflection ...

---

## Referee Comment (RC2) · Anonymous Referee #2 · 10 Nov 2017

The Atmospheric Radiative Transfer Simulator (ARTS) is a versatile radiative transfer tool (apparently) mainly oriented towards various remote sensing applications. The paper by Buehler *et al.* describes version 2.2 of ARTS, with emphasis on new features not present in previous versions. A documentation of ARTS in an open-access journal like GMD would be very welcome, especially as the two peer-reviewed publications describing earlier versions of ARTS as a whole (Buehler et al. (2005a), Eriksson et al. (2011a); papers cited in the manuscript) have been published in JQSRT — which is a respectable journal but not accessible to everyone (including me, as an employee of a national weather service with a very large research department).

[Figure]

While I think it would be important to publish this work, I also think that this paper is clearly in need of improvement. Suggestions for improvement are given below.

**Major comments**

1. The focus on new features of ARTS is certainly a deliberate choice by the authors, but in my opinion it does not work entirely satisfactorily, at least for readers who are not familiar with ARTS beforehand. Much of the discussion is focused on how to compute gaseous line absorption parameters for planets with differing gaseous composition, and on exotic effects like Faraday rotation and Zeeman effect. However, the basic features of ARTS are not described properly (if at all), so the most basic question remains unanswered: *What is ARTS and what can it be used for?*

Therefore, I strongly recommend adding a section that summarizes the basic features of ARTS, not only the new features. The list of issues that should be reported includes at least the following:

- How is the radiative transfer equation solved: (i) scalar or vector radiative transfer; (ii) methods of treating multiple scattering; (iii) solution geometry?

- What is the wavelength region (that can be) considered?

- Treatment of gaseous absorption: (i) basic method (line-by-line?), (ii) absorbing species (you could possibly refer to Table 2); (iii) sources of line data?

- Treatment of emission (Planck function, local thermodynamic equilibrium)

- Treatment of scattering and absorption by particles (e.g., those in clouds)? E.g., what are the choices available for particle single-scattering properties?

- Treatment of the surface?

[Figure]

I recognize that some of this information is provided in the Introduction and in other sections, but it should be provided in a more coherent and systematic manner. It is not necessary to discuss all these features in-depth (most of the information could probably be put in a table), but the basic information should be provided, to put the new features in the context.

2. An unusual feature of this article is that it contains no results and almost no figures. While this is, of course, related to the nature of the article as a model description paper, it makes the paper rather boring to read. Some practical examples would be helpful. A couple of possibilities (you are welcome to invent more examples yourselves):

- In Sect. 2.1, show some examples of how different gas composition for different planets influences, through line broadening, the absorption spectrum of selected gases (e.g., $CO_2$ absorption on Earth vs. Mars).

- In Sect. 3.3.3, show an example of how the Zeeman effect splits a spectral line into several lines. I would be curious to see if this effect is relevant, for example, for Earth's middle/upper atmosphere.

3. The paper provides no information on if and how users of ARTS can verify whether they are using the model correctly. An ideal solution would be to provide a selection of test cases against which the users can compare their results. If this is impractical in the context of this paper (e.g. considering the length of the paper), it would at least be worth mentioning whether such test cases are provided at *www.radiativetransfer.org*, where ARTS is available.

**Minor comments**

1. p. 5, l. 10: How large are the pressure-dependent frequency shifts typically, e.g. at the surface level on Earth?

2. p. 5, Eq. (2): In the exponent, should $n_{\mathrm{air}}$ be $n_a$? Also, how did you choose the constants 0.25 and 1.5 (you state below Eq. (2) that this equation comes without any claim of general validity, but probably these are not totally arbitrary either)?

3. p. 7, l. 15: What is the wavelength range supported by ARTS? It is said here "up to 3 Thz", which corresponds to 100 $\mu$m, but ARTS has also been used for computing radiative fluxes integrated over the thermal infrared region (which starts at about 4 $\mu$m) (Pincus et al. 2015, paper cited in the manuscript). Is it so that the ARTS spectroscopic database starts at 100 $\mu$m and other databases (=HITRAN?) have to be used at shorter wavelengths? Please clarify.

4. p. 7, l. 29: "...Explicit values have been put where available". It is not obvious what this means. "Directly measured values"?

5. p. 9, Table 2: This lacks some gases (most notably CFCs) relevant for climate change. Is it because of the wavelength range considered?

6. p. 21, l. 11–12: Mention that $CO_2$ line mixing is already available in ARTS2.3? Pincus et al. (2015) (paper cited in the manuscipt) discuss this explicitly.

**Technical corrections**

1. p. 2, l. 1: replace "inside" with "among"

2. p. 3, l. 29: replace "triggered" with "motivated"?

3. p. 9, l. 12–13: this should be "dependence ... on"

4. p. 11, l. 10: this should be "Earth's"

5. p. 11, l. 17: replace "sumbillimeter" with "submillimeter"

6. p. 15, l. 2: replace "underestimation" with "underestimate"

7. p. 18, l. 27: this should be "frequency-dependent"

8. p. 19, l. 27: this should be "variety"

9. p. 20, l. 2: this should be "seamlessly"

10. p. 26, l. 2: The length of Larson (2014) is probably less than 1009 pages.

---

## Author Comment (AC1) · 8 Dec 2017

We thank Lutz Gross and both anonymous reviewers for the constructive and thoughtful comments. Work on implementing the many good suggestions on the presentation is in progress, and we will submit a revised manuscript when we have addressed all comments.

---

## Author Response (AR1)

**Responses to Open Discussion Comments**

We thank Lutz Gross and both anonymous reviewers for the constructive and thoughtful comments. Addressing them has improved the paper substantially, we think.

Comment are in *blue italic lettering*, responses in black.

**Executive Editor Comments by Lutz Gross**

*As explained in https://www.geoscientific-model-development.net/about/manuscript_types.html it is preferred for the code to be uploaded as a supplement or to be made available at a data repository with an associated DOI (digital object identifier) for the exact model version described in the paper. Given the impermanence of websites, it is not encouraged that authors put models on their own website. Authors should consider improving the availability with a more permanent arrangement.*

Since ARTS is a well-established model, that has been used by its diverse user community for more than 15 years, we prefer to continue to distribute it through the ARTS website.

*On the website http://www.radiativetransfer.org/misc/download/stable/2.2/ several versions/releases of the code are available for download. The authors need to explicitly state in the manuscript which version the manuscript is based on and are encouraged to provide the code (possibly as stripped version of the tar-ball) as supplement.*

This is a very good point. The website has been updated to distribute and advertise the exact version that is described in the article, and it also now links directly to the article. The link will be updated after acceptance, of course.

**Reviewer 1 Comments**

*Summary:*
*The article describes the current version of the ARTS package, which is a widely used line-by-line radiative transfer model for the thermal spectral range. The paper is not a complete model description but focuses on new features that have been included after the last ARTS reference publication in 2011. The major extension is a planetary toolbox which enables simulations for the planets Venus, Mars and Jupiter in addition to Earth.*
*The paper is interesting to read, and, as also mentioned in the abstract, it focuses on the major extension, the planetary toolbox, in particular, on how the line-by-line calculations need to be adapted for other planets. My major concern is, that the reader, a potential ARTS user, could think, that this is the main application of ARTS. The introduction summarizes the history of the ARTS development, but afterwards I miss an overview of the features included in ARTS, version 2.2. I think that for example an overview table including the most important ARTS features (not only the new ones) should be added.*

Following this suggestion, we have added a new table to the introduction, summarizing the most important general ARTS features.

*Further it would be nice to include examples of use of the new features, if possible including example input files as supplementary data. The first section should also include something like "basic usage", how the program is installed, how it is executed, etc ...*

We think that kind of information is too technical for the paper. But it is very important, of course. So, motivated by the reviewer comment, we have added a new paragraph in the introduction that describes where this kind of information can be found.

*I have several more specific comments and recommend to publish the paper after the revisions.*
**General Comments:**
*Regarding the planetary toolbox extension, I have general questions: I understood that in detail the absorption lines look different for planetary atmospheres other than Earth because one has to take into account different line broadening parameters and also isotopic ratios are different. It is not clear from the paper how large the resulting differences in the absorption lines actually are.*

We have added a new figure showing the absorption cross-section of the same spectral line for all four different planets, to illustrate the effect of the broadening coefficients. Concerning isotopologue ratios, their impact is better discussed in words, since they simply scale the overall absorption. Such a discussion with numerical examples was also added now.

Additionally, we have added the following comment earlier on: "The consequences of not calculating the broadening correctly can be drastic. In a recent comment, Turbet and Tran (2017) point out that using air instead of the correct $CO_2$ broadening coefficients may lead to an error of 13 K in the surface temperature in climate simulations for early Mars."

*A further question is, whether it is possible to estimate the error, which would be obtained for broad-band simulations, when the radiance is integrated over a large number of spectral lines. In particular IR instruments often measure broader bands. Does the error for integrated radiance become larger or smaller than for individual lines? Can ARTS without adaptions for other planets be used to simulate IR broad-band observations of other planets?*

The ESA study that funded the planetary toolbox was limited in scope to frequencies up to 3 THz. The separate broadening data for different gases currently therefore is only available for that range. It would be very desirable to extend it through the thermal IR, but this requires funding for a spectroscopy expert to review the available data, as for the lower frequencies.

So, to answer the question: No, with the current ARTS version it is not very meaningful to simulate broad band IR observations for other planets. The broadening parameters for different gases differ systematically, so most likely the differences will not average out for broadband simulations.

*Specific comments:*
*- Abstract: Please motivate, why you have included Faraday rotation and Zeeman splitting.*

*Why are these features needed for atmospheric radiative transfer?*

The paragraph now reads: "Several other new features are also described, notably radio link budgets (including the effect of Faraday rotation that changes the polarisation state), back-scattering radar simulations, and the treatment of Zeeman splitting. The latter is needed in particular to simulate oxygen lines, for example the ones observed by the various operational microwave satellite temperature sensors of the Advanced Microwave Sounding Unit (AMSU) family."

*p.1., l.16: "computer codes for energy flux computations and remote sensing simulations have been developed in parallel..." - Is this really true? Traditional solution methods like the discrete ordinate method (theory by Chandrasekhar 1950, included e.g. in DISORT code) compute radiance and irradiance simultaneously. I guess that there are more codes that calculate both, e.g. also libRadtran gives radiances, and can also compute (spectrally integrated) irradiances.*

The text was reformulated to make a less strong statement and explain it better: "...From the early days on, high-level computer codes for energy flux computation and remote sensing simulations have developed somewhat independently, and not many complex codes can be used for both applications. Notable exceptions are libRadtran (Emde et al., 2016), which can be used for sensor simulation and flux calculation in the shortwave, and the family of models by AER (Atmospheric and Environmental Research, Clough et al., 2005). The tendency for models to specialise is often not driven by physics (for example low level solvers like DISORT (Stamnes et al., 1988) are suitable for both applications), it rather seems to be driven by practical constraints, resulting from the requirements of the two communities."

*p.5, Eq. 2: Why has this equation been chosen to calculate the pressure shift in ARTS. Where do the constants 0.25 and 1.5 come from?*

The text was expanded at bit, to read: "The origin of these values for our model is in Pumphrey and Buehler (2000), which in turn refers to Pickett (1980), but that paper, although it does discuss the theory of the pressure shift temperature dependence, does not give any explicit value suggestions for the exponents. Despite its shortcomings, we decided to keep the expression for continuity, and in lack of a better one."

*Sec. 2.1: Could you include an example plot showing the difference between the two approaches to calculate line broadening and line shift for a spectral line in the Earth atmosphere? How significant is the difference?*

For Earth the practical difference is really small. We have adjusted the text to make this clearer, and added a new paragraph: "However, the practical difference that this makes for Earth is really small. For the example of the 119 GHz oxygen line, the water vapor broadening parameter is roughly 12% larger than the nitrogen broadening parameter (and the oxygen or self- broadening parameter is quite similar to the nitrogen one). Assuming a water vapor VMR of 1% then increases the total width of the line by only about 0.13%. The reason for the small impact is that there is so much more nitrogen and oxygen which dominates the broadening."

*Here it would also be nice to include ARTS input files (as supplement), showing how to set up the classical absorption calculation and what is changed for the new approach.*

See above. ARTS comes with a large set of example controlfiles, and we now refer to this in the paper. It is in our view better to keep the controlfiles with the code, rather than with the article. It is more user-friendly and easier to maintain in this way.

*p7, l14ff: The line catalogue generated for planetary atmospheres includes data up to 3THz. Are there plans to extend it to the full IR region and possibly also to visible spectral region?*

Not currently, although we would really like to do this. It would require funding for a spectroscopy expert to carefully review the available literature values for the different gases and spectral regions, as for the up to 3 THz region.

Note that HITRAN itself is also moving towards adding separate broadening parameters for different foreign gases, at the moment $H_2$, He and $CO_2$, as mentioned in the paper at the end of Section 2.2. We plan to adopt ARTS to use these extended HITRAN parameters as they become robustly available.

*p.7 l.21: The sources of data for the planetary database are given in Mendrok et al. 2014. The reference is a technical report for ESA, thus it might be difficult to access it. Please include the data sources in the paper (a table would be nice, perhaps references could be included in Table 2).*

There are a lot of different sources, and also the report describes in detail the rationale for the selection, so we think it really cannot be replaced by a simple table. But we completely agree that it is important that the report is accessible. It is freely downloadable from our web page, but the reference did not include the explicit link, which is now fixed.

*p.11, Sec. 2.3, Refractivity: A plot showing refractive index profiles of all planets in the planetary toolbox would be interesting.*

This has been added now.

*p.11, l. 26. Refer to section 3.1 so that reader knows that radio link simulations are actually possible with ARTS and thus the electron contribution to the refractivity is important to include in the model.*

Done.

*Sec. 2.4: Can you show an example (e.g. a specific water vapor line) and demonstrate how this line changes with different isotopic ratios? I.e. plot the same line for Venus, Earth, Mars and Jupiter.*

We think this is better explained in words than by a figure, since isotopologue abundance simply scales the absorption. The following text has been added to address this comment:

"For spectral lines belonging to molecules that contain heavy hydrogen or nitrogen atoms, the change in absorption due to these abundance differences can be very significant. To give an example, the isotopologue abundance of HDO is more than 100 times that of Earth on Venus, 5 times that of Earth on Mars, and only less than 0.2 that of Earth on Jupiter. Because absorption is proportional to abundance, these differences translate directly into absorption differences for spectral lines belonging to this species."

*Sec. 2.5: "Doppler shifts" are mentioned here but at this point the reader does not know about this feature in ARTS. Please refer to Sec. 3.3.4.*

Done.

*Sec. 3.1: Please provide one example for a radio link budget simulation.*

A new figure and associated text were added for this.

*p. 14, l. 10: "considers effects covered by geometrical optics ..." - which effects, please describe more precisely. Do you mean Snell's law of refraction or something else? What is meant with "multi-pathing", please explain briefly.*

The text was expanded to hopefully make this clearer: "The most critical step of these calculations is to establish the propagation path between transmitter and receiver. This step is so far only handled by a quite simple and time consuming algorithm (Eriksson et al., 2011c) and only considers effects covered by geometrical optics. Snell's law is used to determine the bending of the radiation as it travels through the atmosphere, and the algorithm looks for a path that connects transmitter and receiver. In reality, there can be more than one possible path for atmospheres with strong vertical temperature gradients (so called multi-pathing), but this is currently not treated in ARTS. The algorithm simply finds a link path, or determines that no path is possible, due to interception by the planet's surface."

*p. 14, l.15/16: Is it correct that in one of the equations for the "free space loss" the denominator includes 4*pi and in the other (4*pi)ˆ2?*

Yes, the definitions are correct, and differ by a factor of lambda/(4*pi). We reformulated the text to make this clearer and added a references for the Friis formula.

*p. 14, l.29: Please explain in a few words what is "Faraday rotation" and why it has to be considered for radio link budget simulations.*

At this place in the text, we just included a reference to the Section where Faraday rotation is discussed. That section was also revised, to hopefully be clearer.

*Sec. 3.2: Please provide one example for a radar simulation.*

In the meantime we have done a lot of work on the radar module in the current development version, and fixed several issues with the implementation in the version that is the subject of the article. We have therefore decided to not advertise the radar

capability of that version, and instead plan to describe it in a separate article, based on the newer implementation. The complete (short) radar section was therefore removed, and also all references to it.

*p. 15, l.9: Please give an example when T_h and T_a are not equal. How relevant is this inequality for radar simulations?*

This was in the removed radar part, so it is no longer applicable.

*Sec. 3.1: Please provide an example for non-particle polarization. It would be very interesting to show the circular polarisation by Zeeman splitting. How large is the degree of circular polarisation? For which remote sensing applications is this important?*

The Zeeman effect is discussed in much more depth in the two articles that we cite (Larsson et al. (2014) and Larsson (2014)), including many examples. We prefer to not go into more detail on it here, since we fear that the this aspect is too specialized for the general overview article that this aims to be.

*Sec. 3.3.6: I do not understand this section: Absence of attenuation means vacuum, right? In vacuum there is no refraction. How can the radiance change in absence of attenuation? Please explain.*

Absorption and refraction are different processes, though related. Good optical glass has a high refractivity, but (almost no) absorption.

We now write "This law says that, even in the absence of attenuation, the radiance would change along the propagation path due to refraction effects." Perhaps the "would" makes the meaning a bit clearer.

*p. 19, l.9: The "vacuum speed" is the "speed of light", right?*

Yes. We now write "speed of light in vacuum" to be clear.

*Sec.4. The summary and conclusions section is very short. It should include once more a general overview of what is ARTS, including LBL-absorption calculations, RT in thermal range (microwave, IR), scattering by cloud droplets and (oriented) ice crystals, surface reflection ...*

We think that this repetition would have little added value for the reader, so we prefer to keep the conclusions section very short. It was slightly rewritten, however, to concentrate on the most important points.

**Reviewer 2 Comments**

*The Atmospheric Radiative Transfer Simulator (ARTS) is a versatile radiative transfer tool (apparently) mainly oriented towards various remote sensing applications. The paper by Buehler et al. describes version 2.2 of ARTS, with emphasis on new features not present in previous versions. A documentation of ARTS in an open-access journal like GMD would be very welcome, especially as the two peer-reviewed publications describing earlier versions*

*of ARTS as a whole (Buehler et al. (2005a), Eriksson et al. (2011a); papers cited in the manuscript) have been published in JQSRT — which is a respectable journal but not accessible to everyone (including me, as an employee of a national weather service with a very large research department).*

*While I think it would be important to publish this work, I also think that this paper is clearly in need of improvement. Suggestions for improvement are given below.*

***Major comments***

*1. The focus on new features of ARTS is certainly a deliberate choice by the authors, but in my opinion it does not work entirely satisfactorily, at least for readers who are not familiar with ARTS beforehand. Much of the discussion is focused on how to compute gaseous line absorption parameters for planets with differing gaseous composition, and on exotic effects like Faraday rotation and Zeeman effect. However, the basic features of ARTS are not described properly (if at all), so the most basic question remains unanswered: What is ARTS and what can it be used for?*

*Therefore, I strongly recommend adding a section that summarizes the basic features of ARTS, not only the new features.*

The introduction was extended with a brief discussion of general ARTS features, including a feature table, as suggested by the reviewer below.

*The list of issues that should be reported includes at least the following:*
*• How is the radiative transfer equation solved: (i) scalar or vector radiative transfer; (ii) methods of treating multiple scattering; (iii) solution geometry?*

Done (in table).

*• What is the wavelength region (that can be) considered?*

Done (in table).

*• Treatment of gaseous absorption: (i) basic method (line-by-line?), (ii) absorbing species (you could possibly refer to Table 2); (iii) sources of line data?*

Done (in table).

*• Treatment of emission (Planck function, local thermodynamic equilibrium)*

Done (in table).

*• Treatment of scattering and absorption by particles (e.g., those in clouds)? E.g., what are the choices available for particle single-scattering properties?*

Done (in table).

*• Treatment of the surface?*

Done (in table).

*I recognize that some of this information is provided in the Introduction and in other sections, but it should be provided in a more coherent and systematic manner. It is not*

*necessary to discuss all these features in-depth (most of the information could probably be put in a table), but the basic information should be provided, to put the new features in the context.*

We like the suggestion of a feature table, and this is the way we have implemented the general overview now.

*2. An unusual feature of this article is that it contains no results and almost no figures. While this is, of course, related to the nature of the article as a model description paper, it makes the paper rather boring to read. Some practical examples would be helpful. A couple of possibilities (you are welcome to invent more examples yourselves):*

Following the suggestion, we have added five new figures to the paper now.

*• In Sect. 2.1, show some examples of how different gas composition for different planets influences, through line broadening, the absorption spectrum of selected gases (e.g., CO2 absorption on Earth vs. Mars).*

Done, we show the effect of planetary composition on the broadening of an H2O line in a new figure.

*• In Sect. 3.3.3, show an example of how the Zeeman effect splits a spectral line into several lines. I would be curious to see if this effect is relevant, for example, for Earth's middle/upper atmosphere.*

The Zeeman splitting is discussed at length and with figures in the cited articles, so we do not want to duplicate that here.  The effect is important on Earth for example for high-altitude AMSU-A temperature sounding channels (in the 60 GHz O2 line cluster).

*3. The paper provides no information on if and how users of ARTS can verify whether they are using the model correctly. An ideal solution would be to provide a selection of test cases against which the users can compare their results. If this is impractical in the context of this paper (e.g. considering the length of the paper), it would at least be worth mentioning whether such test cases are provided at www.radiativetransfer.org, where ARTS is available.*

We have indeed such a mechanism.  The following text has been added in the introduction to describe it: "…, the distribution includes a large set of sample controlfiles for ARTS that contain predefined setups for various remote sensing instruments, and demonstration cases for various ARTS features. There also is a build target 'make check' that runs a selection of the included controlfiles and compares their computation results against reference data. For the user, this allows to verify that the model works correctly. For the developers, perhaps even more importantly, it helps to ensure continuity and prevents unintentional changes in model output due to source code changes."

*Minor comments*
*1. p. 5, l. 10: How large are the pressure-dependent frequency shifts typically, e.g. at the surface level on Earth?*

For the 183 GHz $H_2O$ line, the "air" shift parameter is -891 Hz/Pa, so at 1000 hPa the shift is almost 90 MHz. Due to the strong broadening, the line center is hard to observe directly at these high pressures, so the main effect of the shift is that it make the line asymmetric.

*2. p. 5, Eq. (2): In the exponent, should nair be na?*

Yes, thanks for spotting the typo.

*Also, how did you choose the constants 0.25 and 1.5 (you state below Eq. (2) that this equation comes without any claim of general validity, but probably these are not totally arbitrary either)?*

We added a bit more explanatory text. There really is only a very poor basis for these constants, and we have now formulated this more strongly. We just use them for lack of better ones.

*3. p. 7, l. 15: What is the wavelength range supported by ARTS? It is said here "up to 3 Thz", which corresponds to 100 μm, but ARTS has also been used for computing radiative fluxes integrated over the thermal infrared region (which starts at about 4 μm) (Pincus et al. 2015, paper cited in the manuscript). Is it so that the ARTS spectroscopic database starts at 100 μm and other databases (=HITRAN?) have to be used at shorter wavelengths? Please clarify.*

Yes, it is only the new planetary line database that is limited to frequencies below 3 THz. This is now stated explicitly in the feature table.

*4. p. 7, l. 29: "...Explicit values have been put where available". It is not obvious what this means. "Directly measured values"?*

The term explicit was misleading. The paragraph was reformulated, to now read: "Foreign species specific line parameters have been derived by a careful literature investigation searching for experimental or theoretical studies specifically devoted to line broadening and shift by He, H2, CO2, or H2. Furthermore, air broadening and shift parameters reported in the HITRAN database are often deduced from individually determined and reported N2 and O2 broadening and shift data. In such cases, we applied the original N2 and O2 literature data in our catalogue compilation. For some combinations of gas species, absorption line and perturbing gas, the broadening and line shift parameters are absent in the literature, simply because spectroscopic studies dealing with these line parameters were never performed. In this case, the values quoted in our catalogue have been reasonably estimated, where the estimation strategy could differ from one absorption species to the other."

*5. p. 9, Table 2: This lacks some gases (most notably CFCs) relevant for climate change. Is it because of the wavelength range considered?*

CFCs (and in general gases that are included in HITRAN only as absorption cross section data, not line parameters) are missing in the ARTS version described. We are working on adding them in the current development version, and also plan a short article about it.

*6. p. 21, l. 11–12: Mention that CO2 line mixing is already available in ARTS2.3? Pincus et al. (2015) (paper cited in the manuscipt) discuss this explicitly.*

We want to avoid too many references to the current development version. Mostly because we want to thoroughly test these new features before they are advertised too widely.

*Technical corrections*
*1. p. 2, l. 1: replace "inside" with "among"*

Done.

*2. p. 3, l. 29: replace "triggered" with "motivated"?*

Done.

*3. p. 9, l. 12–13: this should be "dependence … on"*

Done.

*4. p. 11, l. 10: this should be "Earth's"*

Done.

*5. p. 11, l. 17: replace "sumbillimeter" with "submillimeter"*

Done. (Although we actually like the term "*sumbillimeter*")

*6. p. 15, l. 2: replace "underestimation" with "underestimate"*

This section has been removed now.

*7. p. 18, l. 27: this should be "frequency-dependent"*

Done.

*8. p. 19, l. 27: this should be "variety"*

Done.

*9. p. 20, l. 2: this should be "seamlessly"*

Done.

*10. p. 26, l. 2: The length of Larson (2014) is probably less than 1009 pages.*

Corrected (and also added missing DOI number).

[revised manuscript text omitted]